# ACTIVE LEARNING IN BAYESIAN NEURAL NETWORKS WITH BALANCED ENTROPY LEARNING PRINCIPLE

**Jae Oh Woo**
Samsung SDS Research America
San Jose, CA 95134
`jaeoh.woo@aya.yale.edu`

## ABSTRACT

Acquiring labeled data is challenging in many machine learning applications with limited budgets. Active learning gives a procedure to select the most informative data points and improve data efficiency by reducing the cost of labeling. The info-max learning principle maximizing mutual information such as BALD has been successful and widely adapted in various active learning applications. However, this pool-based specific objective inherently introduces a redundant selection and further requires a high computational cost for batch selection. In this paper, we design and propose a new uncertainty measure, Balanced Entropy Acquisition (BalEntAcq), which captures the information balance between the uncertainty of underlying softmax probability and the label variable. To do this, we approximate each marginal distribution by Beta distribution. Beta approximation enables us to formulate BalEntAcq as a ratio between an augmented entropy and the marginalized joint entropy. The closed-form expression of BalEntAcq facilitates parallelization by estimating two parameters in each marginal Beta distribution. BalEntAcq is a purely standalone measure without requiring any relational computations with other data points. Nevertheless, BalEntAcq captures a well-diversified selection near the decision boundary with a margin, unlike other existing uncertainty measures such as BALD, Entropy, or Mean Standard Deviation (MeanSD). Finally, we demonstrate that our balanced entropy learning principle with BalEntAcq[1] consistently outperforms well-known linearly scalable active learning methods, including a recently proposed PowerBALD, a simple but diversified version of BALD, by showing experimental results obtained from MNIST, CIFAR-100, SVHN, and TinyImageNet datasets.

## 1 INTRODUCTION

Acquiring labeled data is challenging in many machine learning applications with limited budgets. As the dataset size gets bigger and bigger for training a complex model, labeling data by humans becomes more expensive. Active learning gives a procedure to select the most informative data points and improve data efficiency by reducing the cost of labeling.

The active learning problem is well-aligned with a subset selection problem that can find the most efficient but minimal subset from the data pool (Hochbaum, 1996; Nemhauser et al., 1978; Dvoretzky, 1961; Milman, 1971; Spielman & Teng, 2014; Spielman & Woo, 2009; Batson et al., 2009; Spielman & Srivastava, 2011). The difference is that active learning is typically an iterative process where a model is trained and a collection of data points is selected to be labeled from an unlabelled data pool.

It is well-known that any active learning method cannot improve the label complexity better than passive learning (random acquisition) in general (Vapnik & Chervonenkis, 1974; Kääriäinen, 2006; Castro & Nowak, 2008). Under some conditions on labels or models, it is possible to achieve exponential savings (Balcan et al., 2007; Hanneke, 2007; Dasgupta et al., 2005; Hsu, 2010; Dekel et al., 2012; Hanneke, 2014; Zhang & Chaudhuri, 2014; Krishnamurthy et al., 2017; Shekhar et al., 2021; Puchkin & Zhivotovskiy, 2021). Zhu & Nowak (2022b;a) recently proposed a provably

---

[1]Code is available. `https://github.com/jaeohwoo/BalancedEntropy`

exponentially efficient active learning algorithm with abstention with high probability but limited in binary classification cases. On the other hand, although numerous practically successful active learning methods have been proposed, no algorithm has proven efficient enough and linearly scalable to guarantee exponential label savings in general. Therefore, it is still theoretically challenging but important to improve data efficiency significantly.

It is now commonly accepted that standard deep learning models do not capture model uncertainty correctly. The simple predictive probabilities are usually erroneously described as model confidence (Hein et al., 2019). So there is a risk that a model can be misdirecting its outputs with high confidence. However, the predictive distribution generated from Bayesian deep learning models better captures the uncertainty from the data (Gal & Ghahramani, 2016; Kristiadi et al., 2020; Mukhoti et al., 2021; Daxberger et al., 2021). Therefore, we focus on developing an active learning framework in the Bayesian deep neural network model by leveraging the Monte-Carlo (MC) dropout method as a proxy of the Gaussian process (Gal & Ghahramani, 2016) which may facilitate further analysis.

## 1.1 OUR CONTRIBUTIONS

Our proposed active learning method is well-aligned with Bayesian experimental design (Verdinelli & Kadane, 1992; Cohn et al., 1996; Sebastiani & Wynn, 2000; Malinin & Gales, 2018; Foster et al., 2019) with an assumption that the forward active learning iterative process follows the Bayesian prior-posterior framework. Furthermore, our approach is also aligned with Bayesian uncertainty quantification methods (Houlsby et al., 2011; Kandasamy et al., 2015; Kampffmeyer et al., 2016; Gal & Ghahramani, 2016; Alex Kendall & Cipolla, 2017; Gal et al., 2017; Kirsch et al., 2019; Mukhoti et al., 2021; Kirsch et al., 2021) with an assumption that the working neural network model is a Bayesian network (Koller & Friedman, 2009).

In this paper, we extend and improve recent advances in both aspects of Bayesian experimental design and Bayesian uncertainty quantification. We investigate the generalized notion of the joint entropy between model parameters and the predictive outputs by leveraging a point process entropy (McFadden, 1965; Fritz, 1973; Papangelou, 1978; Daley & Vere-Jones, 2007; Baccelli & Woo, 2016). By approximating the marginals using Beta distributions, we then derive an explicit formula of the marginalized joint entropy by estimating Beta parameters from Bayesian deep learning models. As a Bayesian experiment, we revisit the well-known entropy and mutual information measures given expected cross-entropy loss. We show that well-known acquisition measures are functions of marginal distributions through analytical formulas. We propose a new uncertainty measure, Balanced Entropy Acquisition (BalEntAcq), which captures the information balance between the uncertainty of underlying softmax probability and the label variable. Finally, we demonstrate that our balanced entropy learning principle with BalEntAcq consistently outperforms well-known linearly scalable active learning methods, including a recently proposed PowerBALD (Kirsch et al., 2021) for mitigating the redundant selection in BALD (Gal et al., 2017), by showing experimental results obtained from MNIST, CIFAR-100, SVHN, and TinyImageNet datasets.

## 2 BACKGROUND

### 2.1 PROBLEM FORMULATION

We write an unlabeled dataset $\mathcal{D}_{\textbf{pool}}$ and the labeled training set $\mathcal{D}_{\textbf{training}} \subseteq \mathcal{D}_{\textbf{pool}}$ in each active learning iteration. We denote by $\mathcal{D}_{\textbf{training}}^{(n)}$ if it's necessary to indicate the specific $n$-th iteration step. Given $\mathcal{D}_{\textbf{training}}$, we train a Bayesian deep neural network model $\Phi$ with model parameters $\omega \sim \mathbf{p}\left(\omega\right)$.

Then for a data point $\mathbf{x}$ given $\mathcal{D}_{\textbf{training}}$, the Bayesian deep neural network $\Phi$ produces the prediction probability: $\Phi\left(\mathbf{x}, \omega\right) := (P_1(\mathbf{x}, \omega), \cdots, P_C(\mathbf{x}, \omega)) \in \Delta^C$ where $\Delta^C = \{(p_1, \cdots, p_C) : p_1 + \cdots + p_C = 1, p_i \geq 0 \text{ for each } i\}$ and $C$ is the number of classes. For the final class output $Y$, it is assumed to be a multinoulli distribution (or categorical distribution):

$$Y(\mathbf{x}, \omega) := \begin{cases} 1 & \text{with probability } P_1(\mathbf{x}, \omega) \\ \vdots & \vdots \\ C & \text{with probability } P_C(\mathbf{x}, \omega). \end{cases} \tag{1}$$

For the sake of brevity, we sometimes omit $\mathbf{x}$ or $\omega$ by writing $\Phi(\omega)$, $P_i(\omega)$, $Y(\omega)$ or $\Phi$, $P_i$, $Y$ unless we need further clarifications on each data point $\mathbf{x}$. Under this formulation, the oracle (active learning algorithm) selects a subset of data points to add to the next training set, i.e. at $(n+1)$-th iteration, the training set is determined by $\mathcal{D}_{\text{training}}^{(n+1)} = \mathcal{D}_{\text{training}}^{(n)} \cup \{\text{Next training batch from Oracle}\}$. Once the next training batch is selected, the selected batch will be labeled. This means that the ground truth label information of the selected data is added in training set $\mathcal{D}_{\text{training}}^{(n+1)}$ in the next round. Then the goal in active learning is to minimize the number of selected data points to reach a certain level of prediction accuracy.

## 2.2 EXAMPLES OF UNCERTAINTY BASED ACTIVE LEARNING METHODS

In this section, we list up well-known uncertainty measures suitable for Bayesian active learning.

1. **Random**: $\text{Rand}[\mathbf{x}] := U(\omega')$ where $U(\cdot)$ is a uniform distribution which is independent to $\omega$. Random acquisition function assigns a random uniform value on $[0, 1]$ to each data point.

2. **BALD** (Bayesian active learning by disagreement) (Lindley, 1956; Houlsby et al., 2011; Gal et al., 2017): $\text{BALD}[\mathbf{x}] := \mathfrak{I}(\omega, Y(\mathbf{x}, \omega))$, where $\mathfrak{I}(\cdot, \cdot)$ represents a mutual information between random measures. BALD captures the mutual information between the model parameters and the predictive output of the data point. In practice, we calculate the mutual information between the predictive output and the predictive probabilities.

3. **Entropy** (Shannon, 1948): $\text{Ent}[\mathbf{x}] := -\sum_i (\mathbb{E}P_i) \log (\mathbb{E}P_i)$. Entropy is the Shannon entropy with respect to the expected predictive probability. Entropy can be the uncertainty of the prediction probability. Moreover, under the cross-entropy loss, we may also interpret the entropy measure as an expected loss gain since $-\log(\mathbb{E}P_i)$ is the cross-entropy loss given the ground truth label is the class $i$.

4. **Mean standard deviation (MeanSD)** (Cohn et al., 1996; Kampffmeyer et al., 2016; Alex Kendall & Cipolla, 2017): $\text{MeanSD}[\mathbf{x}] := \frac{1}{C} \sum_i \sqrt{\mathbb{E}P_i^2 - (\mathbb{E}P_i)^2}$. Mean standard deviation captures the average of the standard deviations for each marginal distribution.

5. **PowerBALD** (Farquhar et al., 2021; Kirsch et al., 2021): $\text{PowerBALD}[\mathbf{x}] := \log \text{BALD}[\mathbf{x}] + Z$, where $Z$ is an independently generated random value from Gumbel distribution with the location $\mu = 0$ and the scale $\beta = 1$ parameters, see Wikipedia (2023). We use $\beta = 1$ as a default choice suggested by Kirsch et al. (2021). The motivation of this randomized acquisition is to mitigate the redundant selection by diversifying selected multi-batch points. In general, we do not know which parameter $\beta$ will be the optimal choice.

In a multiple acquisition scenario, we simply add the above uncertainty values for each data point $\mathbf{x}_i$:

$$\text{AcqFunc}[\mathbf{x}_1, \cdots, \mathbf{x}_n] := \sum_{i=1}^{n} \text{AcqFunc}[\mathbf{x}_i], \tag{2}$$

where $\text{AcqFunc} \in \{\text{Rand}, \text{BALD}, \text{Ent}, \text{MeanSD}, \text{PowerBALD}\}$.

## 2.3 SUMMARY OF OTHER ACTIVE LEARNING APPROACHES

Cohn et al. (1996) provided one of the first statistical analyses in active learning, establishing how to synthesize queries that reduce the model's forward-looking error by minimizing its variance leveraging MacKay's closed-form variance approximation (MacKay, 1992). In this fashion, there exists a line of works in Bayesian experimental design (Chaloner & Verdinelli, 1995; Lindley, 1956; Verdinelli & Kadane, 1992; Cohn et al., 1996; Sebastiani & Wynn, 2000; Roy & McCallum, 2001; Yoon et al., 2013; Vincent & Rainforth, 2017; Foster et al., 2019; 2021; Jha et al., 2022) with an assumption that the forward active learning iterative process follows Bayesian prior-posterior framework.

On the other hand, in active learning, accommodating both the information uncertainty and the diversification of the acquired samples is essential to improve the performance under multi-batch acquisition scenarios. In a theoretical perspective, the most natural way to combine the uncertainty and the diversification seems to leverage reasonable sub-modular functions, e.g. Nearest neighbor set

function (Wei et al., 2015), BatchBALD (Kirsch et al., 2019), Determinantal Point Process (Bıyık et al., 2019) and SIMILAR (Kothawade et al., 2021) with sub-modular information measures, and then/or apply a fast linear-time algorithm to find a diversified multi-batch with a provable performance guarantee (Nemhauser & Wolsey, 1978; Nemhauser et al., 1978; Ene & Nguyen, 2017; Yaroslavtsev et al., 2020; Schreiber et al., 2020; Iyer et al., 2021a;b; Li et al., 2022). Although a fast linear-time solver is available for general sub-modular functions, there still exists a gap with practical implementation, such as high memory requirements, which makes the computation unscalable for identifying multi-batch acquisition points, e.g., BatchBALD (Kirsch et al., 2019). Similar to the sub-modular function optimization, there exist many customized optimization approaches, e.g. CoreSet (Sener & Savarese, 2018) and more approaches (Guo, 2010; Joshi et al., 2010; Elhamifar et al., 2013; Yang et al., 2015; Wang & Ye, 2015).

Another recent approach is to look at parameters of the neural network and to diversify points such as BADGE (Ash et al., 2020) with gradients and BAIT (Ash et al., 2021) with Fisher information. There also exist network architectural design focused approaches such as Learning loss by designing loss prediction layers (Yoo & Kweon, 2019), UncertainGCN and CoreGCN (Caramalau et al., 2021) with graph neural networks , VAAL (Sinha et al., 2019) and TA-VAAL (Kim et al., 2021) by applying adversarial learning methods.

## 3   BAYESIAN NEURAL NETWORK MODEL

We adopt the Bayesian neural network framework introduced in Gal & Ghahramani (2016). The core idea in the Bayesian neural network is leveraging the MC dropout feature to generate a distribution of the predictive probability as an output at inference time. Under mild assumptions, it turns out that it is equivalent to an approximation to a Gaussian Process (Rasmussen & Williams, 2006; Neal, 1996; Williams, 1997; Gal & Ghahramani, 2016; Lee et al., 2017).

### 3.1   SOFTMAX PROBABILITY MARGINAL APPROXIMATELY FOLLOWS BETA DISTRIBUTION

We may consider a Bayesian neural network model $\Phi$ as a random measure, i.e., stochastic process parametrized by $\mathcal{D}_{\textbf{training}}$ over the data set $\mathcal{D}_{\textbf{pool}}$. Given a data point $\mathbf{x} \in \mathcal{D}_{\textbf{pool}}$, $\Phi\left(\mathbf{x}, \omega\right)$ produces a random probability distribution in a simplex $\Delta^C$. This analogy has a close connection with the construction of random discrete distribution, originally introduced by Kingman (1975). Since then, random measure construction has been extensively developed in Bayesian nonparametrics, and it is well-known that Dirichlet probability having Beta marginals plays the central role in the construction of the random discrete distribution (Kingman, 1977; Ferguson, 1973; Pitman & Yor, 1997; Pitman et al., 2002; Broderick et al., 2012; Orlitsky et al., 2004; Santhanam et al., 2014). It is the main motivation of the Beta distribution approximation. Many kinds of literature similarly assume the Dirichlet distribution after the softmax in the Bayesian neural network.

As illustrated by Milios et al. (2018), we may follow the construction of Dirichlet distribution. Following the approach by Ferguson (1973), a Dirichlet probability can be constructed through a collection of independent Gamma distributions. On the other hand, each marginal in Gaussian Process (approximated by Bayesian neural network) in the softmax output having dependent components follows a log-normal distribution (before the normalization, but after the exponentiation in softmax). Then by applying the shape similarity between a log-normal distribution and Gamma distribution, the construction of random probability from log-normal distributions would produce an approximated Dirichlet distribution. Therefore we may assume that the marginal distribution would approximately follow the Beta distribution.

Alternatively, as an analytical approach, we may see that Beta approximation can be justified through Laplace approximation (MacKay, 1998; Hennig et al., 2012; Hobbhahn et al., 2020; Daxberger et al., 2021). There exists a mapping between multivariate Gaussian distribution and Dirichlet distribution under a softmax basis. Then Beta distribution follows as a marginal distribution of Dirichlet distribution. Therefore we may assume that Beta approximation exists through Laplace approximation under the assumption that the Bayesian neural network produces the multivariate Gaussian distribution (as a marginalized Gaussian process over finite rank covariate function) before the softmax layer (Neal, 1996; Williams, 1997; Gal & Ghahramani, 2016; Lee et al., 2017).

In practice, once we estimate the sample mean and sample variance for each marginal of $\Phi(\mathbf{x}, \omega)$, we can estimate two parameters of the Beta distribution as follows. Assume that $P_i \sim \text{Beta}(\alpha_i, \beta_i)$. If $\mathbb{E}P_i = m_i$ and $\text{Var}P_i = \sigma_i^2$, then $\alpha_i = \frac{m_i^2(1-m_i)}{\sigma_i^2} - m_i, \quad \beta_i = \left(\frac{1}{m_i} - 1\right)\alpha_i$. When $P_i \sim \text{Beta}(\alpha_i, \beta_i)$, $\mathbb{E}P_i = \frac{\alpha_i}{\alpha_i + \beta_i} = m$ and $\text{Var}P_i = \frac{\alpha_i \beta_i}{(\alpha_i + \beta_i)^2(\alpha_i + \beta_i + 1)} = \sigma_i^2$. Solving the equation with respect to $\alpha_i$ and $\beta_i$, then the equation follows.

## 3.2 MARGINALIZED JOINT ENTROPY IN BAYESIAN NEURAL NETWORK

Assume that each $P_i \sim \text{Beta}(\alpha_i, \beta_i)$ by applying Beta approximation. We may define a quantity of the marginalized joint entropy (See Appendix A.1 and A.2) and we find an equivalent formulation as follows:

$$\text{MJEnt}[\mathbf{x}] := -\sum_i \mathbb{E}_{P_i}\left[P_i \log\left(P_i f(P_i)\right)\right] = \underbrace{\sum_i (\mathbb{E}P_i) h(P_i^+)}_{\text{posterior uncertainty}} + \underbrace{\mathfrak{I}(\omega, Y)}_{\text{epistemic uncertainty}} + \underbrace{\mathbb{E}_\omega\left[H\left(Y|\omega\right)\right]}_{\text{aleatoric uncertainty}}.$$

(3)

where $h(\cdot)$ is a differential entropy, $\mathfrak{I}(\cdot, \cdot)$ represents mutual information between two quantities, $H(\cdot)$ is Shannon entropy, $P_i^+$ is the conjugate Beta posterior entropy of $P_i$ which follows $P_i^+ \sim \text{Beta}(\alpha_i + 1, \beta_i)$. We call the first term in equation 3 to be the posterior uncertainty. We may interpret the posterior uncertainty as an expected posterior entropy assuming that we observed a positive sample of the class toward $P_i$ for each $i$ without knowing the true class label. The first term is always non-positive, and is maximized (equals to 0) when each $P_i^+$ is $\text{Beta}(1, 1)$, i.e., Uniform on $[0, 1]$. So $-\infty < \text{MJEnt}[\mathbf{x}] \leq H(Y)$, where we note that $H(Y) = \mathfrak{I}(\omega, Y) + \mathbb{E}_\omega\left[H\left(Y|\omega\right)\right]$.

The epistemic uncertainty captures the model uncertainty (as BALD), and the aleatoric uncertainty captures the data uncertainty (Matthies, 2007). Therefore the marginalized joint entropy, $\text{MJEnt}[\mathbf{x}]$ is a decomposition of three types of uncertainty values.

## 3.3 ENTROPY IS FOR MAXIMIZING AN EXPECTED CROSS-ENTROPY LOSS

Given a ground-truth label $\{Y = i\}$, the cross-entropy loss of the neural network can be given as $\text{loss}\left(\Phi(\mathbf{x}, \omega), Y = i\right) = -\log \mathbb{E}P_i$. Therefore, inspired by the expected loss in risk management (Jorion, 2000), we can calculate the expected cross-entropy loss without knowing the truth label:

$$\text{ExpectedLoss}[\mathbf{x}] := \sum_{i=1}^C \mathbb{P}\left[Y = i\right] \text{loss}\left(\Phi(\mathbf{x}, \omega), Y = i\right) = -\sum_i (\mathbb{E}P_i) \log(\mathbb{E}P_i) = \text{Ent}[\mathbf{x}].$$

Based on the re-formulation, we may interpret that entropy acquisition is for maximizing an expected cross-entropy loss in a selection of acquisition points, aligning the idea with the learning loss (Yoo & Kweon, 2019). The natural question is, "Once we acquire a data point that maximizes entropy acquisition, can we remove/or learn this expected cross-entropy amount of loss at the future stage of the active learning?". The answer could be "No." The exhaustive loss acquisition could only happen if the neural network perfectly over-fits the training data. Therefore, there exists a gap between a realistic neural network training scenario and the objective of the entropy acquisition. Our equivalent loss interpretation without introducing epistemic or aleatoric uncertainty confirms typical perceptions of why the entropy acquisition might not be successful in practice, even in the single-point acquisition scenario.

## 3.4 BALD IS A FUNCTION OF MARGINALS AND IS STRONGLY ALIGNED WITH MAXIMIZING AN EXPECTED CROSS-ENTROPY LOSS DIFFERENCE UPTO THE NEXT ITERATION

We have the mutual information between $\omega$ and $Y$ and it is the same as the mutual information between the encoded message and the channel output since $Y$ depends only on $\Phi(\mathbf{x}, \omega)$ (Gal et al., 2017), $\text{BALD}[\mathbf{x}] := \mathfrak{I}(\omega, Y) = \mathfrak{I}(\Phi(\mathbf{x}, \omega), Y(\mathbf{x}, \omega))$. By assuming that $\Phi(\mathbf{x}, \omega)$ follows a Dirichlet distribution, we can calculate the mutual information analytically (Woo, 2022). Then by investigating further into the analytical mutual information formula, we see that the marginal distributions $P_i$'s in $\Phi(\mathbf{x}, \omega)$ are sufficient to estimate BALD. This marginal representation is an important phenomenon as dependency of all coordinates could be removed at the predictive stage (Bobkov & Madiman,

2011, See Conjecture V.5). Therefore we can represent BALD through Beta marginal distributions as follows. See Appendix A.4 for more details.

**Theorem 3.1.** *Under Beta marginal distribution approximation, let $P_i \sim Beta(\alpha_i, \beta_i)$ in $\Phi(\mathbf{x}, \omega)$. Then the mutual information BALD[$\mathbf{x}$] can be estimated as follows:*

$$BetaMarginalBALD[\mathbf{x}] := \sum_{i=1}^{C} (\alpha_i - 1) \Psi(\alpha_i + \beta_i) - \sum_{i=1}^{C} \left(\frac{\alpha_i}{\alpha_i + \beta_i}\right) \log\left(\frac{\alpha_i}{\alpha_i + \beta_i}\right) - \sum_{i=1}^{C} \frac{\alpha_i(\alpha_i - 1)}{\alpha_i + \beta_i} \Psi(\alpha_i)$$

$$- \sum_{i=1}^{C} \frac{\beta_i(\alpha_i - 1)}{\alpha_i + \beta_i} \Psi(\alpha_i + \beta_i + 1) + \sum_{i=1}^{C} \left(\frac{\alpha_i^2}{\alpha_i + \beta_i}\right) [\Psi(\alpha_i + 1) - \Psi(\alpha_i + \beta_i + 1)].$$

As a Bayesian experimental design process, we may assume that each Beta marginal distribution $P_i$ with the ground-truth label $\{Y = i\}$ of the next trained model would follow the Beta posterior distribution $P_i^+$. Without this assumption, existing choices of acquisition functions such as BALD or MeanSD might not be well-justified. For example, what is the implication of maximizing mutual information through the active learning process with a Bayesian neural network? How is it different from the maximization of the entropy acquisition? To answer these questions, leveraging our Beta marginalization and considering the similar idea of expected information gain (Foster et al., 2019), we may consider the expected cross-entropy loss difference between the current stage model and the next stage model.

$$ExpectedEffectiveLoss[\mathbf{x}] := \sum_{i=1}^{C} \mathbb{E}P_i \left[-\log \mathbb{E}P_i - \left(-\log \mathbb{E}P_i^+\right)\right] = \sum_{i=1}^{C} \left(\frac{\alpha_i}{\alpha_i + \beta_i}\right) \left[\log\left(\frac{\alpha_i + 1}{\alpha_i + \beta_i + 1}\right) - \log\left(\frac{\alpha_i}{\alpha_i + \beta_i}\right)\right].$$

ExpectedEffectiveLoss captures the effective amount of cross-entropy loss for the model to learn after the acquisition. By definition, we see that ExpectedEffectiveLoss aims to exclude the undesirable over-fitting scenario assumption unlike Entropy acquisition.

Since Digamma function $\Psi(x) \sim \log x - \frac{1}{2x}$ where $f(x) \sim g(x)$ implies $\lim_{x \to \infty} f(x)/g(x) = 1$, we may expect that BetaMarginalBALD[$\mathbf{x}$] and ExpectedEffectiveLoss[$\mathbf{x}$] would behave similarly. Figure 1 shows the Spearman's rank correlations among different acquisition measures upto a class dimension $C = 10,000$. We observe that BetaMarginalBALD behaves equally like the original BALD and we confirm that BALD and MeanSD are strongly aligned with maximizing ExpectedEffectiveLoss. Therefore, acquiring points through BALD or MeanSD could be a better strategy than Entropy because BALD or MeanSD takes into account the effective loss acquisition instead of the unrealistic full amount of the loss acquisition.

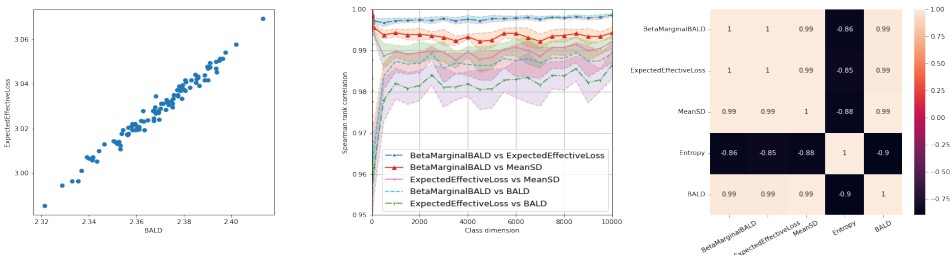

Figure 1: Scatter plot at $C = 10,000$ between BALD and ExpectedEffectiveLoss (left), Spearman's rank correlations over various class dimensions (middle), and Spearman's rank correlation matrix at $C = 10,000$ (right). The relationship between BetaMarginalBALD and ExpectedEffectiveLoss consistently captures a high rank-correlation with $> 99.6\%$ regardless of the class dimensions. BALD and ExpectedEffectiveLoss show $> 97.5\%$ rank-correlation. We randomly generate 100 softmax applied $C$-dimensional Gaussian samples and repeated the process 10 times. Shaded band shows the standard deviation.

## 4    BALANCED ENTROPY LEARNING PRINCIPLE

The previous section shows that well-known acquisition measures have an objective toward cross-entropy loss and are closely related to marginal distributions. However, according to Farquhar et al.

(2021), to be successful in active learning, they hypothesize that it is crucial to find a good balance between active learning bias and over-fitting bias under over-parametrized neural networks. Although they claim that it might not be possible to achieve the ultimate active learning goal without having the full information, we may still define the balanced entropy (BalEnt) to be a ratio between the marginalized joint entropy equation 3 and the augmented entropy:

$$\text{BalEnt}[\mathbf{x}] := \frac{\text{MJEnt}[\mathbf{x}]}{\text{Ent}[\mathbf{x}] + \log 2} = \frac{\sum_i (\mathbb{E}P_i) h(P_i^+) + H(Y)}{H(Y) + \log 2}. \tag{4}$$

Recall that we call the first term in MJEnt[$\mathbf{x}$] to be posterior uncertainty, and it is an expected posterior entropy of underlying marginals. BalEnt captures the information balance between the posterior uncertainty from the model $\Phi$ and entropy of the label variable $Y$.

## 4.1 IMPLICATIONS OF BALANCED ENTROPY

To understand the implication of BalEnt[$\mathbf{x}$], we can prove the following Theorem 4.1.

**Theorem 4.1.** *Let $\Delta^{-1} := \lfloor 2e^{H(Y)} \rfloor$ and $\Upsilon := \{I_n\}$, a collection of evenly divided intervals in $[0, 1]$ where $I_n := \left[(n-1)\Delta, n\Delta\right)$ for $n = 1, \cdots, (\Delta^{-1} - 1)$ and $I_{\Delta^{-1}} := [1 - \Delta, 1]$. Let $\bar{P}_i$ be a discretized random variable over $\Upsilon$ of $P_i$ from $\Phi(\mathbf{x}, \omega)$. For any estimator $\hat{P}_i$ of $\bar{P}_i$ given the label $\{Y = i\}$ we have*

$$\mathbb{E}\left[\mathbb{P}\left[\hat{P}_i \neq \bar{P}_i \middle| Y = i\right]\right] \geq \frac{\sum_i (\mathbb{E}P_i) h(P_i^+) + H(Y)}{H(Y) + \log 2}(1 + \varepsilon_1) - \varepsilon_2 = \text{BalEnt}[\mathbf{x}](1 + \varepsilon_1) - \varepsilon_2,$$

*where $\varepsilon_1, \varepsilon_2 \geq 0$ are adjustment terms depending on $\Delta$ such that $\varepsilon_1 \to 0$ and $\varepsilon_2 \to 0$ as $\Delta \to 0$.*

Theorem 4.1 tries to answer the following inverse problem. For the unlabeled data point, $\mathbf{x}$, if we know the information of the label $\{Y = i\}$, how much can we reliably estimate the underlying probability $P_i$ from the model $\Phi$? As we know that $-\log P_i$ is the cross-entropy loss of the trained model with $Y$, it equivalently answers the estimation error probability of the loss prediction under a unit precision up to $-\log \Delta$ level. For the precision level, we are assuming to carry $-\log \Delta \approx H(Y) + \log 2$ nats - natural unit of information, re-scaled amount of bits, matching the enumerator with MJEnt[$\mathbf{x}$] term. It is not clear how to determine a better choice of the precision level $-\log \Delta$. But we may understand the denominator $H(Y) + \log 2$ is for normalizing the term BalEnt[$\mathbf{x}$] $\leq 1$ as a probability. Then the sign of BalEnt[$\mathbf{x}$] becomes very important. BalEnt[$\mathbf{x}$] $\geq 0$ implies that it could be impossible to perfectly predict the loss $-\log P_i$ given currently available information. i.e., there could exist information imbalance between the model and the label approximately starting from BalEnt[$\mathbf{x}$] $= 0$. Therefore, insight from Theorem 4.1 suggests us a new direction for our main active learning principle. We define our primary acquisition function, namely, balanced entropy learning acquisition (BalEntAcq), as follows:

$$\text{BalEntAcq}[\mathbf{x}] := \begin{cases} \text{BalEnt}[\mathbf{x}]^{-1} & \text{if BalEnt}[\mathbf{x}] \geq 0, \\ \text{BalEnt}[\mathbf{x}] & \text{if BalEnt}[\mathbf{x}] < 0, \end{cases}$$

Since the information imbalance exists at least from BalEnt[$\mathbf{x}$] $= 0$, we prioritize to fill the information gap from BalEnt[$\mathbf{x}$] $= 0$ toward positively increasing direction which aligns with choosing the entropy increasing contours. If we try to fill the information imbalance gap from the highest BalEnt[$\mathbf{x}$], the information imbalance would still exist around BalEnt[$\mathbf{x}$] $= 0$ area. Therefore, it might not improve the active learning performance much. See Appendix A.13.2 and A.13.3 for different prioritization and precision level results. That's the motivation why we take the reciprocal of BalEnt[$\mathbf{x}$] when BalEnt[$\mathbf{x}$] $\geq 0$.

## 4.2 TOY EXAMPLE ILLUSTRATION

To illustrate the behavior of BalEntAcq and its relationship with other uncertainty measures, we train a simple Bayesian neural network with a 3-class moon dataset in $\mathbb{R}^2$. Then we calculate each acquisition measure for all fixed lattice points in the square domain by assuming that the unlabeled pool is highly regularized (or uniform). i.e., by evenly discretizing the domain, we obtain each uncertainty value for each lattice point. The total number of lattice points is around 0.6 million.

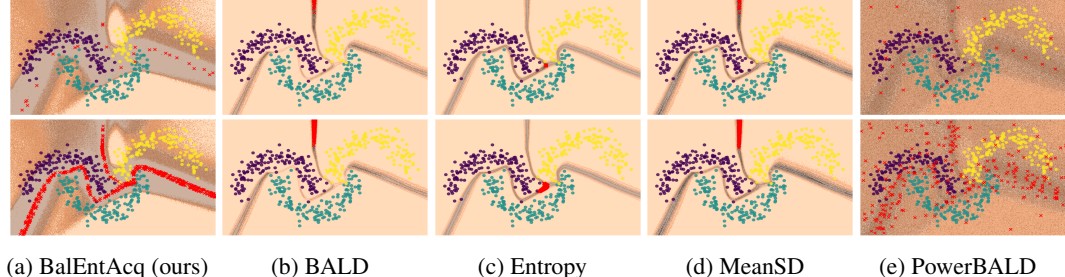

| (a) BalEntAcq (ours) | (b) BALD | (c) Entropy | (d) MeanSD | (e) PowerBALD |

Figure 2: Top-$K$ selected points are marked by red color. The first row shows the top $K = 25$ points. The second row shows the top $K = 500$ point selections among around 0.6 million grid points.

Then we choose top-$K$ high uncertainty values for each method to observe the prioritized region for each method. We use $K = 25$ and $K = 500$. Figure 2 illustrates the top-$K$ points selected by each method. The most significant phenomenon is that BalEntAcq's selection is highly diversified near the decision boundary showing a bifurcated margin because we are prioritizing the surface area of $\{\text{BalEnt}[\mathbf{x}] \geq 0\}$. This is well-aligned with the strategy avoiding high aleatoric points. (See Appendix A.11) Then we can imagine to conduct a uniform sampling on each contour surface $\{\text{BalEnt}[\mathbf{x}] = \lambda\}$ for each $\lambda \geq 0$, as we move to the surface for each $\lambda < 0$. That's why we observe bifurcated but diversified and balanced selection near the decision boundary with BalEngAcq in Figure 2-(a) when $K = 25$. On the other hand, there is a preferred area for each method from other measures except PowerBALD. PowerBALD shows a good diversification, but it could select non-informative points.

## 5 EXPERIMENTAL RESULTS

In this section, we demonstrate the performance of BalEntAcq from MNIST (LeCun & Cortes, 2010), CIFAR-100 (Krizhevsky et al., 2012), SVHN (Netzer et al., 2011), and TinyImageNet (Le & Yang, 2015) datasets under various scenarios. We used a single NVIDIA A100 GPU for each experiment, and details about the experiments are explained in Appendix A.13. We test Random, BALD, Entropy, MeanSD, PowerBALD, and BalEntAcq measures. We add BADGE for additional baseline. Note that all acquisition measures except BADGE in our experiments are standalone quantities, so they can be easily parallelized, i.e., linearly scalable.

**Single acquisition active learning with MNIST.** MNIST is the most popular and elementary dataset to validate the performance of image-based deep learning models initially. We use a simple convolutional neural network (CNN) model applying dropouts to all layers with a single acquisition size. The primary purpose of this single acquisition experiment is to validate our proposed balanced entropy approach by removing the contribution of diversification unlike multi-batch acquisition scenario.

**Fixed features with CIFAR-100 and 3×CIFAR-100.** In recent years, significant efforts have been made on building an efficient framework of unsupervised or self-supervised feature learning such as SimCLR (Chen et al., 2020a;b), MoCo (He et al., 2020), BYOL (Grill et al., 2020), SwAV (Caron et al., 2020), DINO (Caron et al., 2021), etc. As an application in active learning, we may leverage the feature space from the unsupervised feature learning without explicitly knowing true labels but construct a good representation space. In our experiments, we adopt SimCLR for simplicity with ResNet-50 to build a feature space for CIFAR-100.

With 3×CIFAR-100 dataset, we observe the effect of the redundant information treatment for each method by adding three identical points. We use the same fixed feature obtained from SimCLR with CIFAR-100. We may observe how each method effectively diversifies the selection under a redundant data pool scenario by fixing the feature space.

**Pre-trained backbone with SVHN and strong data augmentation with TinyImageNet.** In this experiment, we follow a typical image classification scenario in practice. We use the ResNet-18 backbone for SVHN and the ResNet-50 backbone for TinyImageNet with ImageNet pre-trained model for model architecture, and the last linear classification layer is replaced with a simple Bayesian

neural network with dropouts. In TinyImageNet iterations, we re-use the previously trained model for the next training. So the pre-trained ImageNet weight is only used at the initial iteration. We apply strong data augmentations for TinyImageNet, including random crop, random flip, random color jitter, and random grayscale. Under this scenario, the feature space from the backbone is continuously evolving and keeps confused as the training and active learning process proceeds. Because of the strong data augmentation and batch normalization in ResNet-18 or ResNet-50, the decision boundary keeps confused, implying that the Bayesian experimental design assumption might not hold. However, we still want to observe the general behavior of each measure and how to improve the accuracy under a more dynamic feature space.

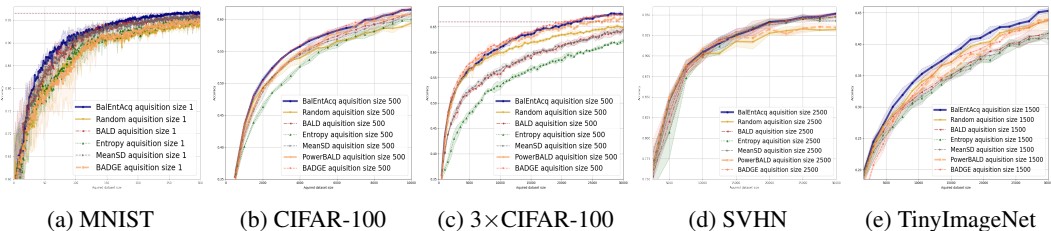

| (a) MNIST | (b) CIFAR-100 | (c) 3×CIFAR-100 | (d) SVHN | (e) TinyImageNet |

Figure 3: Active learning accuracy curves obtained from various scenarios. Our proposed BalEntAcq outperforms well-known acquisition measures, and we repeated the experiment 3 times.

Table 1: Selected accuracy table. Mean and standard deviation are from 3 repeated experiments.

| Scenario | Full dropouts + CNN | | | Fixed feature | | Redundant images + Fixed feature | | Backbone | | Backbone + Augmentation | |
|---|---|---|---|---|---|---|---|---|---|---|---|
| Dataset/Acq. Size/Test size | MNIST/1/10,000 | | | CIFAR-100/500/10,000 | | 3×CIFAR-100/500/10,000 | | SVHN/2,500/26,032 | | TinyImageNet/1,500/10,000 | |
| Train Size/Pool Size | 50/60,000 | 100/60,000 | 300/60,000 | 5,000/50,000 | 10,000/50,000 | 15,000/150,000 | 30,000/150,000 | 15,000/73,257 | 30,000/73,257 | 15,000/100,000 | 30,000/100,000 |
| Random | 78.6 ± 4.9% | 86.4 ± 2.7% | 93.6 ± 0.7% | 55.5 ± 0.4% | 59.4 ± 0.5% | 61.9 ± 0.2% | 64.9 ± 0.3% | 91.8 ± 0.6% | 93.2 ± 0.2% | 37.1 ± 0.3% | 43.8 ± 0.1% |
| BALD | 82.6 ± 1.3% | 90.5 ± 0.8% | 95.3 ± 0.4% | 56.2 ± 0.5% | 60.8 ± 0.3% | 58.8 ± 0.2% | 64.6 ± 0.6% | 92.5 ± 0.8% | 94.8 ± 0.2% | 35.2 ± 0.7% | 41.8 ± 0.4% |
| Entropy | 77.4 ± 2.6% | 87.7 ± 2.0% | 94.8 ± 0.3% | 54.9 ± 0.4% | 60.0 ± 0.3% | 56.7 ± 0.8% | 62.3 ± 0.4% | **92.6 ± 0.4%** | 94.8 ± 0.2% | 35.1 ± 0.4% | 41.8 ± 0.4% |
| MeanSD | 83.4 ± 2.2% | 90.6 ± 1.1% | 96.0 ± 0.2% | 56.0 ± 0.1% | 60.9 ± 0.4% | 59.4 ± 0.5% | 64.3 ± 0.3% | 92.5 ± 0.6% | 94.3 ± 0.2% | 34.7 ± 0.4% | 40.9 ± 0.6% |
| PowerBALD | - | - | - | 56.0 ± 0.1% | 60.9 ± 0.1% | 62.8 ± 0.4% | 66.6 ± 0.1% | 92.2 ± 0.6% | 93.5 ± 0.2% | 35.8 ± 0.7% | 43.9 ± 0.8% |
| BADGE (not-scalable) | 77.0 ± 6.1% | 86.5 ± 4.2% | 94.8 ± 0.4% | **57.4 ± 0.1%** | **61.8 ± 0.1%** | **64.0 ± 0.2%** | **67.4 ± 0.1%** | **92.9 ± 0.4%** | 95.0 ± 0.3% | 37.2 ± 0.6% | 43.9 ± 0.3% |
| **BalEntAcq (ours)** | **85.4 ± 1.0%** | **91.4 ± 1.3%** | **96.5 ± 0.1%** | **57.2 ± 0.2%** | **61.5 ± 0.2%** | **63.5 ± 0.5%** | **67.4 ± 0.1%** | **92.5 ± 0.8%** | **95.2 ± 0.1%** | **38.5 ± 0.2%** | **45.3 ± 0.4%** |

**Discussion.** BalEntAcq consistently outperforms other linearly scalable baselines in all datasets, as shown in Table 1. BADGE performs similarly to Entropy under a single acquisition scenario in MNIST because BADGE focuses on maximizing the loss gradient similar to Entropy, as we explained in Section 3.3. BADGE shows better performances at first when we fix the feature space, but our BalEntAcq eventually catches up with the performance of BADGE. We also note that BADGE is not a linearly scalable method. Under dynamic feature scenarios in SVHN or TinyImageNet, we observe that our BalEntAcq performs better. Considering the acquisition calculation time (see Appendix A.18), our BalEntAcq should be a better choice. Figure 3 shows the full active learning curves. For CIFAR-100 and 3×CIFAR-100 cases, by fixing features, we control/remove all other effects possibly affecting the model's performance, such as data augmentation or the role of backbone in the classification. As demonstrated in Figure 2, BalEntAcq is very efficient in selecting diversified points along the decision boundary. Instead, PowerBALD suffers from improving accuracy because it focuses more on diversification/randomization by missing the information near the decision boundary. For SVHN or TinyImageNet, BalEntAcq shows better performance again. We suppose that diversification near the decision boundary in BalEntAcq also plays the data exploration because the representation space keeps evolving with the backbone training.

## 6 CONCLUSION

In this paper, we designed and proposed a new uncertainty measure, Balanced Entropy Acquisition (BalEntAcq), which captures the information balance between the underlying probability and the label variable through Beta approximation with a Bayesian neural network. BalEntAcq offers a diversified selection and is unique compared to other uncertainty measures. We expect that our proposed balanced entropy measure does not have to be confined to active learning problems in general. BalEntAcq would improve the diversified selection process in many other Bayesian frameworks. Therefore, we look forward to having further follow-up studies with broad applications beyond the active learning problems.

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

# A APPENDIX

## A.1 BAYESIAN NEURAL NETWORK AND MUTUAL INFORMATION

We may formulate the Bayesian neural network $\Phi$ as a well-known encoder-decoder framework. The sender sends a message $(\mathbf{x}, \omega)$ with a random key $\omega$ through the Bayesian neural network, then the receiver receives a message $Y(\mathbf{x}, \omega)$. Figure 4 illustrates a diagram in this process.

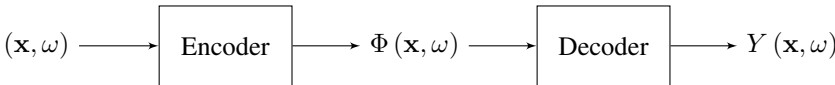

Figure 4: Bayesian neural network encoder-decoder framework

Under this framework, controlling $\omega$ is difficult, but we can control the family of the encoded messages $\Phi(\mathbf{x}, \omega) := (P_1(\mathbf{x}, \omega), \cdots, P_C(\mathbf{x}, \omega))$ in a tractable manner (Gal et al., 2017; Kingma & Welling, 2014; Tzikas et al., 2008). We can easily prove that the mutual information between $\omega$ and $Y$ is the same as the mutual information between the encoded $\Phi(\mathbf{x}, \omega)$ and the predictive output $Y$ since $Y$ depends only on $\Phi(\mathbf{x}, \omega)$:

$$\mathrm{BALD}[\mathbf{x}] := \Im(\omega, Y(\mathbf{x}, \omega)) = H(Y(\mathbf{x}, \omega)) - \mathbb{E}_\omega[H(Y(\mathbf{x}, \omega)|\omega)] \tag{5}$$

$$= H(Y(\mathbf{x}, \omega)) - \mathbb{E}_\Phi[H(Y(\mathbf{x}, \omega)|\Phi(\mathbf{x}, \omega))] = \Im(\Phi(\mathbf{x}, \omega), Y(\mathbf{x}, \omega)), \tag{6}$$

where $H(Y(\mathbf{x}, \omega))$ represents the Shannon entropy by marginalizing out the randomness of $\omega$ in $Y(\mathbf{x}, \omega)$ and $\Im(\cdot, \cdot)$ represents a mutual information between two quantities.

The formulations of the mutual information equation 5 - equation 6 look natural, but we need to note that $\omega$ or $\Phi(\mathbf{x}, \omega)$ is on a continuous domain, and $Y(\mathbf{x}, \omega)$ is on a discrete domain. This combined domain implies that we cannot directly apply Shannon entropy and differential entropy notions (Cover, 1999). One immediate question is what the joint entropy between $\Phi(\mathbf{x}, \omega)$ and $Y(\mathbf{x}, \omega)$ is. For this, we can leverage point process entropy (McFadden, 1965; Fritz, 1973; Papangelou, 1978; Daley & Vere-Jones, 2007; Baccelli & Woo, 2016) by generalizing the notion of the entropy in this combined domain. We consider the joint entropy of $\Phi(\mathbf{x}, \omega)$ and $Y(\mathbf{x}, \omega)$, denoting by $\mathfrak{H}(\Phi(\mathbf{x}, \omega), Y(\mathbf{x}, \omega))$ through the point process entropy. We write a Janossy density function (Daley & Vere-Jones, 2007) $j(\mathbf{p}, y = i)$ of $(\Phi(\mathbf{x}, \omega), Y(\mathbf{x}, \omega))$ on $\Delta^C \times [C]$ as follows:

$$j(\mathbf{p}, y = i) = p_i f(\mathbf{p}), \tag{7}$$

where $\mathbf{p} := (p_1, \cdots, p_C)$ and $f(\cdot)$ is a density function of $\Phi(\mathbf{x}, \omega)$. Then the joint entropy of $\Phi(\mathbf{x}, \omega)$ and $Y(\mathbf{x}, \omega)$ can be defined as

$$\mathfrak{H}(\Phi(\mathbf{x}, \omega), Y(\mathbf{x}, \omega)) = -\sum_{i=1}^C \int_{\Delta^c} j(\mathbf{p}, y = i) \log j(\mathbf{p}, y = i) \, \mathrm{d}\mathbf{p}. \tag{8}$$

By plugging equation 7 into equation 8, we have the following identity.

$$\mathfrak{H}(\Phi(\mathbf{x}, \omega), Y(\mathbf{x}, \omega)) = H(Y(\mathbf{x}, \omega)) + \mathbb{E}_Y[h(\Phi(\mathbf{x}, \omega)|Y(\mathbf{x}, \omega))], \tag{9}$$

where $H(\cdot)$ represents the usual Shannon entropy, and $h(\cdot)$ represents the usual differential entropy. By applying Jensen's inequality, we may derive a marginalized joint entropy as an upper bound of the joint entropy (See Appendix A.2):

$$\mathfrak{H}(\Phi(\mathbf{x}, \omega), Y(\mathbf{x}, \omega)) \leq -\sum_i \mathbb{E}_{P_i}[P_i \log(P_i f(P_i))], \tag{10}$$

where we ambiguously write $f(\cdot)$ to be a density function for each $P_i$.

## A.2 DERIVATION OF MARGINALIZED JOINT ENTROPY WITH THE POINT PROCESS ENTROPY

The Janossy density function resides in a combination of continuous and discrete domains (Daley & Vere-Jones, 2007). For the Janossy density of $(\Phi(\mathbf{x}, \omega), Y(\mathbf{x}, \omega))$ on $\Delta^C \times [C]$, we may follow the

classical approach:

$$\mathbb{P}\left(P_1 \in [p_1 + \mathrm{d}p_1], \cdots, P_C \in [p_c + \mathrm{d}p_c], Y = i\right)$$
$$\approx \mathbb{P}\left(Y = i \middle| P_1 = p_1, \cdots, P_C = p_C\right) \mathbb{P}\left(P_1 \in [p_1 + \mathrm{d}p_1], \cdots, P_C \in [p_c + \mathrm{d}p_c]\right)$$
$$\approx p_i f\left(p_1, \cdots, p_C\right) \mathrm{d}p_1 \cdots \mathrm{d}p_C, \tag{11}$$

where $f(\cdot)$ is a density function of $\Phi(\mathbf{x}, \omega)$. So we may write the Janossy density of $(\Phi(\mathbf{x}, \omega), Y(\mathbf{x}, \omega))$ as follows:

$$j\left(p_1, \cdots, p_C, y = i\right) = p_i f\left(p_1, \cdots, p_C\right). \tag{12}$$

Following the point process entropy (McFadden, 1965; Fritz, 1973; Papangelou, 1978; Daley & Vere-Jones, 2007), the joint entropy of $\Phi(\mathbf{x}, \omega)$ and $Y(\mathbf{x}, \omega)$ can be defined as

$$\mathfrak{H}\left(\Phi(\mathbf{x}, \omega), Y(\mathbf{x}, \omega)\right) = -\sum_{i=1}^{C} \int_{\Delta^c} j\left(p_1, \cdots, p_C, y = i\right) \log j\left(p_1, \cdots, p_C, y = i\right) \mathrm{d}p_1 \cdots \mathrm{d}p_C. \tag{13}$$

We note that

$$\int_{\Delta^c} p_i f\left(p_1, \cdots, p_C\right) \mathrm{d}p_1 \cdots \mathrm{d}p_C = \int_{[0,1]} p_i f\left(p_i\right) \mathrm{d}p_i = \mathbb{E}P_i. \tag{14}$$

We may split the Jannosy density into two pieces:

$$j\left(p_1, \cdots, p_C, y = i\right) = \left(\mathbb{E}P_i\right) \left(\frac{p_i}{\mathbb{E}P_i} f\left(p_1, \cdots, p_C\right)\right). \tag{15}$$

Plugging equation 15 into equation 13, we have

$$\mathfrak{H}\left(\Phi(\mathbf{x}, \omega), Y(\mathbf{x}, \omega)\right) = H\left(Y(\mathbf{x}, \omega)\right) + \mathbb{E}_Y\left[h\left(\Phi(\mathbf{x}, \omega) | Y\right)\right]. \tag{16}$$

On the other hand,

$$(6) = -\sum_{i=1}^{C} \int_{\Delta^c} j\left(p_1, \cdots, p_C, y = i\right) \log j\left(p_1, \cdots, p_C, y = i\right) \mathrm{d}p_1 \cdots \mathrm{d}p_C$$
$$= -\sum_{i=1}^{C} \int_{\Delta^c} \left(\mathbb{E}P_i\right) \left(\frac{p_i}{\mathbb{E}P_i} p\left(p_1, \cdots, p_C\right)\right) \log \left(\mathbb{E}P_i\right) \left(\frac{p_i}{\mathbb{E}P_i} p\left(p_1, \cdots, p_C\right)\right) \mathrm{d}p_1 \cdots \mathrm{d}p_C$$
$$= -\sum_{i=1}^{C} \left(\mathbb{E}P_i\right) \log \left(\mathbb{E}P_i\right) - \sum_{i=1}^{C} \int_{\Delta^c} \left(\mathbb{E}P_i\right) \left(\frac{p_i}{\mathbb{E}P_i} p\left(p_1, \cdots, p_C\right)\right) \log \left(\frac{p_i}{\mathbb{E}P_i} p\left(p_1, \cdots, p_C\right)\right) \mathrm{d}p_1 \cdots \mathrm{d}p_C. \tag{17}$$

We apply Jensen's inequality on the second term (by focusing on each summand). For each $i \in \{1, \cdots C\}$,

$$-\left(\mathbb{E}P_i\right) \int_{\Delta^c} \left(\frac{p_i}{\mathbb{E}P_i} p\left(p_1, \cdots, p_C\right)\right) \log \left(\frac{p_i}{\mathbb{E}P_i} p\left(p_1, \cdots, p_C\right)\right) \mathrm{d}p_1 \cdots \mathrm{d}p_C$$
$$= -\left(\mathbb{E}P_i\right) \int_{p_i} \int_{\Delta^c \setminus \{p_i\}} \left(\frac{p_i}{\mathbb{E}P_i} p\left(p_1, \cdots, p_C\right)\right) \log \left(\frac{p_i}{\mathbb{E}P_i} p\left(p_1, \cdots, p_C\right)\right) \mathrm{d}p_{1 \cdots C}^{-i} \mathrm{d}p_i$$
$$\leq -\left(\mathbb{E}P_i\right) \int_{p_i} \left(\int_{\Delta^c \setminus \{p_i\}} \frac{p_i}{\mathbb{E}P_i} p\left(p_1, \cdots, p_C\right) \mathrm{d}p_{1 \cdots C}^{-i}\right) \log \left(\int_{\Delta^c \setminus \{p_i\}} \frac{p_i}{\mathbb{E}P_i} p\left(p_1, \cdots, p_C\right) \mathrm{d}p_{1 \cdots C}^{-i}\right) \mathrm{d}p_i$$
$$= -\int_{p_i} p_i f(p_i) \log \left(\frac{p_i}{\mathbb{E}P_i} f(p_i)\right) \mathrm{d}p_i = -\mathbb{E}_{P_i} \left[P_i \log \left(\frac{P_i}{\mathbb{E}P_i} f(P_i)\right)\right], \tag{18}$$

where $\mathrm{d}p_{1 \cdots C}^{-i}$ indicates $\mathrm{d}p_1 \cdots \mathrm{d}p_C$ except $\mathrm{d}p_i$. By combining all terms together, we have

$$(6) \leq -\sum_{i=1}^{C} \left(\mathbb{E}P_i\right) \log \left(\mathbb{E}P_i\right) - \sum_{i=1}^{C} \mathbb{E}_{P_i} \left[P_i \log \left(\frac{P_i}{\mathbb{E}P_i} f(P_i)\right)\right] = -\sum_{i} \mathbb{E}_{P_i} \left[P_i \log \left(P_i f(P_i)\right)\right]. \tag{19}$$

## A.3 Equivalent formulation of marginalized joint entropy

Let us assume that $P_i \sim \text{Beta}(\alpha_i, \beta_i)$ and $P_i^+ \sim \text{Beta}(\alpha_i + 1, \beta_i)$.

$$\text{MJEnt}[\mathbf{x}] = -\sum_i \mathbb{E}_{P_i}\left[P_i \log\left(P_i f(P_i)\right)\right] = -\sum_i \int_0^1 p_i f(p_i) \log\left(p_i f(p_i)\right) \mathrm{d}p_i$$

$$= -\sum_i \left(\mathbb{E}P_i\right) \int_0^1 \frac{p_i f(p_i)}{\mathbb{E}P_i} \log\left(\frac{p_i f(p_i)}{\mathbb{E}P_i}\right) \mathrm{d}p_i - \sum_i \left(\mathbb{E}P_i\right) \log\left(\mathbb{E}P_i\right) \tag{20}$$

$$= \sum_i \left(\mathbb{E}P_i\right)\left[h(P_i^+) - \log\left(\mathbb{E}P_i\right)\right], \tag{21}$$

where $h(P_i^+)$ is the differential entropy of $P_i^+$.

## A.4 Proof of Theorem 3.1

Let $\boldsymbol{\eta} = (\eta_1, \cdots, \eta_C)$ and $\boldsymbol{\eta}(i, ++) = (\eta_1, \cdots, \eta_{i-1}, \eta_i + 1, \eta_{i+1}, \cdots, \eta_C)$. Let $B(\boldsymbol{\eta}) = \frac{\Gamma(\eta_1)\cdots\Gamma(\eta_C)}{\Gamma\left(\sum_{k=1}^C \eta_k\right)}$, and $\Gamma(\cdot)$ is a Gamma function. Assume that $\Phi(\mathbf{x}, \omega) := (P_1, \cdots, P_C) \sim$ Dirichlet$(\eta_1, \cdots, \eta_C)$.

**Theorem A.1.** *Woo (2022, Theorem III.1) The analytical formula of the mutual information BALD$[\mathbf{x}]$ is the following.*

$$\textit{DirichletBALD}[\mathbf{x}] := \left(\sum_{k=1}^C \eta_k - C\right) \Psi\left(\sum_{k=1}^C \eta_k\right) - \sum_{i=1}^C (\eta_i - 1) \Psi(\eta_i) - \sum_{i=1}^C \left(\frac{\eta_i}{\sum_{k=1}^C \eta_k}\right) \log\left(\frac{\eta_i}{\sum_{k=1}^C \eta_k}\right)$$

$$+ \sum_{i=1}^C \sum_{j \neq i} \frac{(\eta_j - 1) B(\boldsymbol{\eta}(i, ++))}{B(\boldsymbol{\eta})} \left[\Psi(\eta_j) - \Psi\left(\left(\sum_{k=1}^C \eta_k\right) + 1\right)\right]$$

$$+ \sum_{i=1}^C \frac{\eta_i B(\boldsymbol{\eta}(i, ++))}{B(\boldsymbol{\eta})} \left[\Psi(\eta_i + 1) - \Psi\left(\left(\sum_{k=1}^C \eta_k\right) + 1\right)\right],$$

*where $\Psi(\cdot)$ is a Digamma function.*

Given the above theorem, we can simplify the formula further:

DirichletBALD$[\mathbf{x}]$

$$= \left(\sum_{k=1}^C \eta_k - C\right) \Psi\left(\sum_{k=1}^C \eta_k\right) - \sum_{i=1}^C (\eta_i - 1) \Psi(\eta_i) - \sum_{i=1}^C \left(\frac{\eta_i}{\sum_{k=1}^C \eta_k}\right) \log\left(\frac{\eta_i}{\sum_{k=1}^C \eta_k}\right)$$

$$+ \sum_{i=1}^C \sum_{j \neq i} \left(\frac{\eta_i(\eta_j - 1)}{\sum_{k=1}^C \eta_k}\right) \left[\Psi(\eta_j) - \Psi\left(\left(\sum_{k=1}^C \eta_k\right) + 1\right)\right]$$

$$+ \sum_{i=1}^C \left(\frac{\eta_i^2}{\sum_{k=1}^C \eta_k}\right) \left[\Psi(\eta_i + 1) - \Psi\left(\left(\sum_{k=1}^C \eta_k\right) + 1\right)\right]$$

$$= \left(\sum_{k=1}^C \eta_k - C\right) \Psi\left(\sum_{k=1}^C \eta_k\right) - \sum_{i=1}^C \left(\frac{\eta_i}{\sum_{k=1}^C \eta_k}\right) \log\left(\frac{\eta_i}{\sum_{k=1}^C \eta_k}\right)$$

$$- \sum_{i=1}^C \frac{\eta_i(\eta_i - 1)}{\sum_{k=1}^C \eta_k} \Psi(\eta_i) - \sum_{i=1}^C (\eta_i - 1)\left(1 - \frac{\eta_i}{\sum_{k=1}^C \eta_k}\right) \Psi\left(\left(\sum_{k=1}^C \eta_k\right) + 1\right)$$

$$+ \sum_{i=1}^C \left(\frac{\eta_i^2}{\sum_{k=1}^C \eta_k}\right) \left[\Psi(\eta_i + 1) - \Psi\left(\left(\sum_{k=1}^C \eta_k\right) + 1\right)\right].$$

Therefore DirichletBALD is a function of marginals of $\Phi(\mathbf{x}, \omega)$ with Dirichlet distribution parameters $\eta_i$ and $\sum_{i=1}^C \eta_i$. Under Beta marginal distribution assumption, by letting $\eta_i = \alpha_i$ and $\sum_{k=1}^C \eta_k = \alpha_i + \beta_i$ for any $i$ since each

marginal distribution of Dirichlet distribution follows Beta distribution, we have

$$\text{BetaMarginalBALD}[\mathbf{x}] := \sum_{i=1}^{C} (\alpha_i - 1) \Psi(\alpha_i + \beta_i) - \sum_{i=1}^{C} \left( \frac{\alpha_i}{\alpha_i + \beta_i} \right) \log \left( \frac{\alpha_i}{\alpha_i + \beta_i} \right) - \sum_{i=1}^{C} \frac{\alpha_i (\alpha_i - 1)}{\alpha_i + \beta_i} \Psi(\alpha_i)$$

$$- \sum_{i=1}^{C} \frac{\beta_i (\alpha_i - 1)}{\alpha_i + \beta_i} \Psi(\alpha_i + \beta_i + 1) + \sum_{i=1}^{C} \left( \frac{\alpha_i^2}{\alpha_i + \beta_i} \right) [\Psi(\alpha_i + 1) - \Psi(\alpha_i + \beta_i + 1)].$$

Therefore Theorem 3.1 follows.

On the other hand, since Beta marginal distributions are sufficient to calculate the mutual information, the same idea can be applied to the aleatoric uncertainty.

**Corollary 1.** *Under Beta marginal distribution approximation, let $P_i \sim Beta(\alpha_i, \beta_i)$ in $\Phi(\mathbf{x}, \omega)$. Then the aleatoric uncertainty can be estimated as follows:*

$$BetaMarginalAleatoricUncertainty[\mathbf{x}] := -\sum_{i=1}^{C} (\alpha_i - 1) \Psi(\alpha_i + \beta_i) + \sum_{i=1}^{C} \frac{\alpha_i (\alpha_i - 1)}{\alpha_i + \beta_i} \Psi(\alpha_i)$$

$$+ \sum_{i=1}^{C} \frac{\beta_i (\alpha_i - 1)}{\alpha_i + \beta_i} \Psi(\alpha_i + \beta_i + 1) - \sum_{i=1}^{C} \left( \frac{\alpha_i^2}{\alpha_i + \beta_i} \right) [\Psi(\alpha_i + 1) - \Psi(\alpha_i + \beta_i + 1)].$$

### A.5 PROOF OF THEOREM 4.1

First let a positive integer $\Delta^{-1} > 0$ be given and let $\Upsilon := \{I_n\}$, a collection of evenly divided intervals in $[0, 1]$ where $I_n := [(n-1)\Delta, n\Delta)$ for $n = 1, \cdots, (\Delta^{-1} - 1)$ and $I_{\Delta^{-1}} := [1 - \Delta, 1]$. Let $\bar{P}_i$ be a discretized random variable over $\Upsilon$ of $P_i$ from $\Phi(\mathbf{x}, \omega)$. i.e., $\bar{P}_i = (n - \frac{1}{2}) \Delta$ if $P_i \in I_n$ such that $\mathbb{P}[\bar{P}_i = (n - \frac{1}{2}) \Delta] = \mathbb{P}[P_i \in I_n]$. For any estimator $\hat{P}_i$ of $\bar{P}_i$ given the label $\{Y = i\}$, by applying Fano's inequality (Fano, 1961; Anantharam & Verdu, 1996)(Cover, 1999, Theorem 2.10.1), we have (note that our $\log$ has a base $e$)

$$\mathbb{P}\left[ \hat{P}_i \neq \bar{P}_i \Big| Y = i \right] \geq \frac{H(\bar{P}_i | Y = i) - \log 2}{\log \Delta^{-1}} = \frac{H(\bar{P}_i | Y = i) - \log 2}{-\log \Delta}. \tag{22}$$

We note that Shannon entropy and the differential entropy have the following connection (Cover, 1999, Theorem 8.3.1):

$$H(\bar{P}_i | Y = i) + \log \Delta = h(P_i | Y = i) + \epsilon_i = h(P_i^+) + \epsilon_i, \tag{23}$$

where $\epsilon_i$ is an adjustment constant depending on $\Delta$ such that $\epsilon_i \to 0$ as $\Delta \to 0$. Note that $\epsilon_i$ does not have to be non-negative. Then we can rewrite the inequality as follows:

$$\mathbb{P}\left[ \hat{P}_i \neq \bar{P}_i \Big| Y = i \right] \geq \frac{h(P_i^+) - \log \Delta - \log 2}{-\log \Delta} + \frac{\epsilon_i}{-\log \Delta}. \tag{24}$$

Taking the expectation with respect to $Y$, we have

$$\mathbb{E}\left[ \mathbb{P}\left[ \hat{P}_i \neq \bar{P}_i \Big| Y = i \right] \right] \geq \frac{\sum_i (\mathbb{E}P_i) h(P_i^+) - \log \Delta - \log 2}{-\log \Delta} + \frac{\sum_i (\mathbb{E}P_i) \epsilon_i}{-\log \Delta} =: (**).$$

If we let $\Delta^{-1} = \lfloor 2e^{H(Y)} \rfloor$, there exists a $\delta \geq 0$ such that

$$H(Y) + \log 2 - \delta = -\log \Delta = \log \lfloor 2e^{H(Y)} \rfloor \leq H(Y) + \log 2. \tag{25}$$

We also note that $\delta \to 0$ as $H(Y) \to \infty$ (or equivalently $\Delta \to 0$). Therefore, when $\Delta^{-1} = \lfloor 2e^{H(Y)} \rfloor$, we have

$$(**) = \frac{\sum_i (\mathbb{E}P_i) h(P_i^+) + H(Y) - \delta}{H(Y) + \log 2 - \delta} + \frac{\sum_i (\mathbb{E}P_i) \epsilon_i}{H(Y) + \log 2 - \delta}$$

$$\geq \frac{\sum_i (\mathbb{E}P_i) h(P_i^+) + H(Y)}{H(Y) + \log 2} \left( 1 + \frac{\delta}{H(Y) + \log 2 - \delta} \right) - \frac{\sum_i (\mathbb{E}P_i) |\epsilon_i| + \delta}{H(Y) + \log 2}. \tag{26}$$

Let $\varepsilon_1 = \frac{\delta}{H(Y)+\log 2 - \delta}$ and $\varepsilon_2 = \frac{\sum_i (\mathbb{E}P_i)|\epsilon_i| + \delta}{H(Y)+\log 2} \geq 0$. Since $\epsilon_i \to 0$ and $\delta \to 0$ as $\Delta \to 0$, $\varepsilon_1$ and $\varepsilon_2 \to 0$ as $\Delta \to 0$. Therefore Theorem 4.1 follows.

As a final remark, we note that $\Delta^{-1}$ can be regarded as a variant of the discrete entropy power (Wang et al., 2014; Woo & Madiman, 2015; Madiman et al., 2019; 2021; Haghighatshoar et al., 2014; Jog & Anantharam, 2014).

### A.6 More Beta marginal formulations

With Beta approximation, we are able to describe Beta marginal formulation of MeanSD. Since we are matching the variance of each marginal distribution, the empirical value of MeanSD should be the same as BetaMarginalMeanSD.

$$\text{BetaMarginalMeanSD}[\mathbf{x}] := \frac{1}{C} \sum_{i=1}^{C} \sqrt{\frac{\alpha_i \beta_i}{(\alpha_i + \beta_i)^2 (\alpha_i + \beta_i + 1)}} = \text{MeanSD}[\mathbf{x}]. \qquad (27)$$

In Foster et al. (2019), the expected information gain has been proposed and studied. We may also formulate the expected information gain with Beta marginal distributions.

$$\begin{aligned}
\text{BetaMarginalEIG}[\mathbf{x}] := &H(Y) - \mathbb{E}H\left(Y^+ \middle| Y = i\right) \\
= &\sum_{i=1}^{C} \left(\frac{\alpha_i}{\alpha_i + \beta_i}\right) \left[ \sum_{j=1}^{C} \left(\frac{\alpha_j + \delta_i(j)}{\alpha_j + \beta_j + 1}\right) \log\left(\frac{\alpha_j + \delta_i(j)}{\alpha_j + \beta_j + 1}\right) - \log\left(\frac{\alpha_i}{\alpha_i + \beta_i}\right) \right],
\end{aligned}$$
$$(28)$$

where $Y^+$ is a categorical random variable over the posterior probability given $Y = i$, $\delta_i(j) = 1$ if $i = j$, and $\delta_i(j) = 0$ otherwise.

### A.7 Beta marginal approximation visualization examples

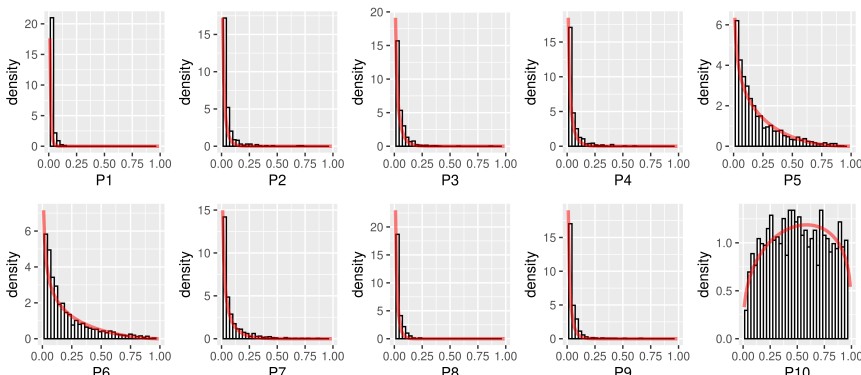

Figure 5: An example of Beta approximations (red lines) for each marginal distribution after applying softmax layer in MNIST dataset. Each Beta distribution is estimated by calculating the sample mean and sample variance of the histogram generated by the Bayesian deep learning model.

Figure 5 and Figure 6 shows an example of Beta approximations obtained from the MNIST and CIFAR-10 datasets. $P_1, \cdots, P_{10}$ show each marginal distribution of the predictive probability of each digit. We observe that the Beta approximation is a reasonable approximation.

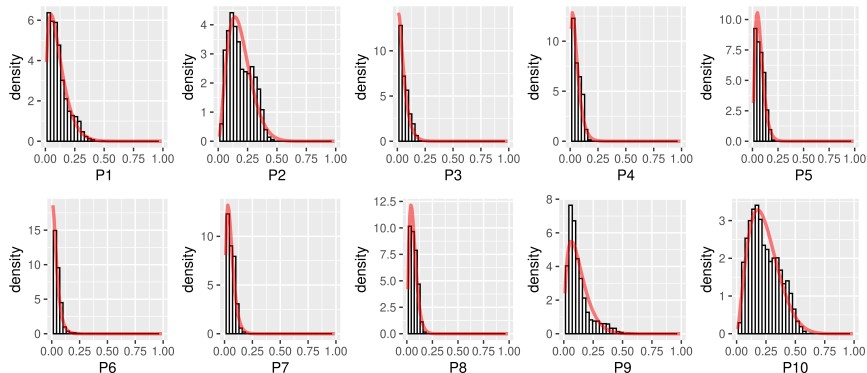

Figure 6: An example of Beta approximations (red lines) for each marginal distribution after applying softmax layer in CIFAR-10 dataset.

## A.8    RANK CORRELATION STUDY WITH BETAMARGINALEIG

A good advantage of explicit formula is that we can study the behavior of each measure directly. For example, if $C = 2$ and $\Phi(\mathbf{x}, \omega) \sim \text{Dirichlet}(\alpha, \beta)$ such that $P_1 \sim \text{Beta}(\alpha, \beta)$ and $P_2 \sim \text{Beta}(\beta, \alpha)$, we are able to plot the behavior of each Beta marginal measure.

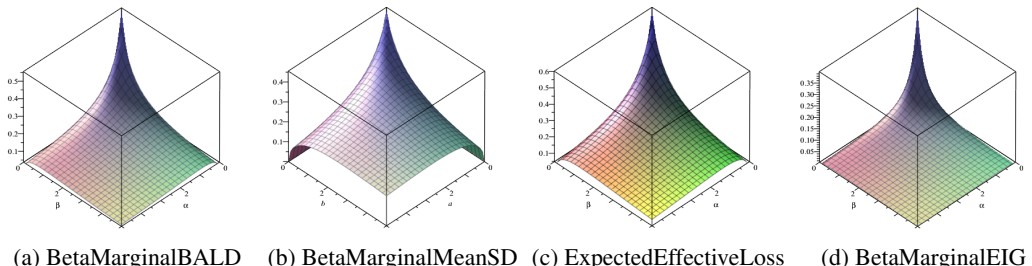

(a) BetaMarginalBALD    (b) BetaMarginalMeanSD    (c) ExpectedEffectiveLoss    (d) BetaMarginalEIG

Figure 7: 3D plot of each uncertainty measure when Beta marginal assumption holds.

With BetaMarginalEIG, we are able to generate the same type of plot shown in Figure 1. EIG shows positive correlations with BALD and MeanSD, but the correlation is around 70% implying that EIG might show more variations.

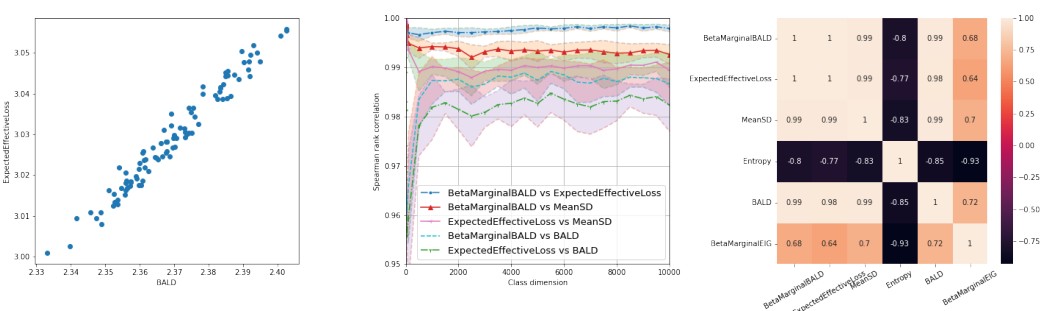

Figure 8: Same experiments with BetaMarginalEIG described in Figure 1 from the main article. This is another independent experiment (as a validation), so the captured correlation values are slightly different.

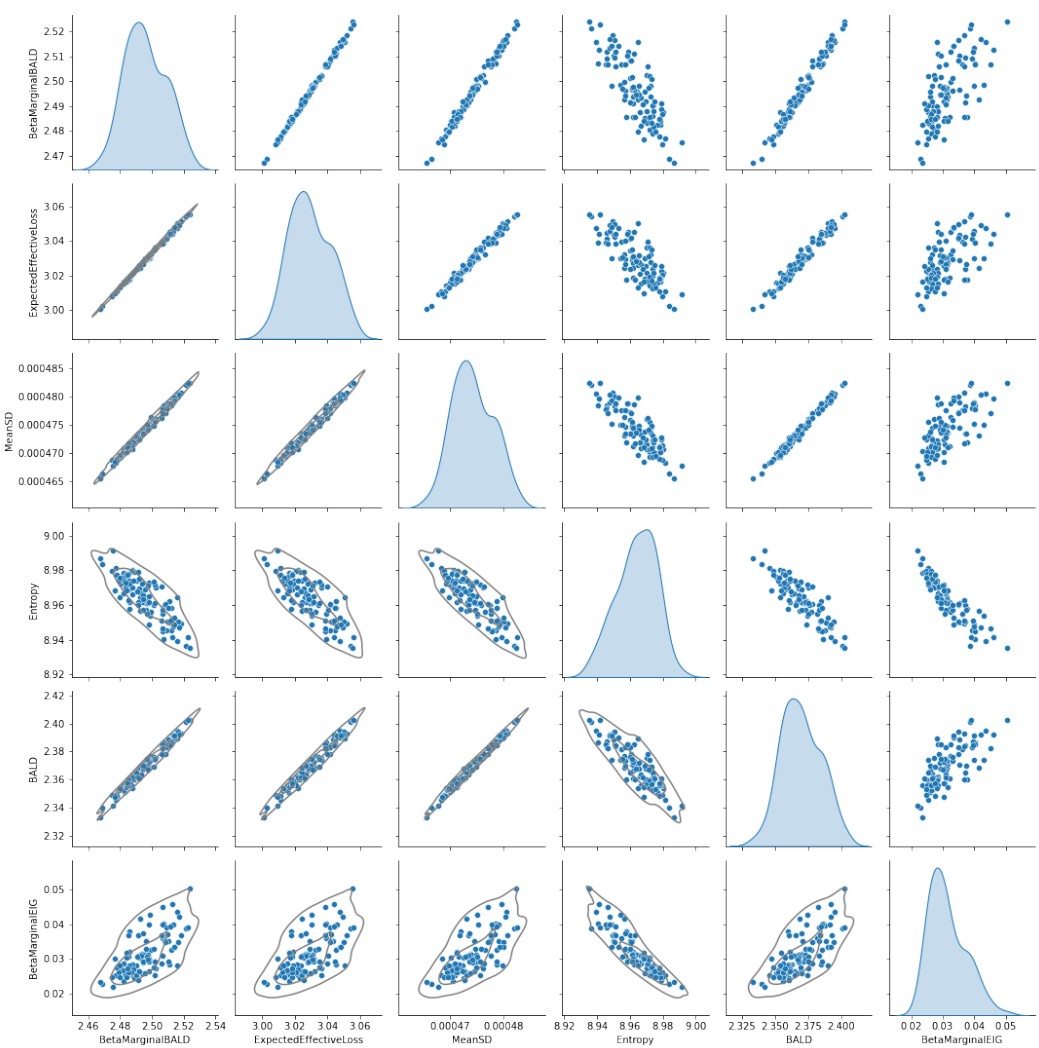

Figure 9: Pairwise scatter plot at $C = 10,000$ companion with Figure 8.

## A.9 BALD AND BETAMARGINALBALD

Although BALD and BetaMarignalBALD shows high rank-correlation (under softmax applied Gaussian distribution assumption), we might wonder how much different they are in the value. We plot the RMSE (rooted mean square error) between two measures. Under Dirichlet distribution assumption, RMSE between BALD and BetaMarginalBALD is $< 0.002$ upto $C \leq 1,000$. However, under softmax-applied Gaussian distribution assumption, RMSE between BALD and BetaMarginalBALD shows $< 0.07$ upto $C \leq 1,000$. This implies that Beta marginal approximation still preserves a high rank-correlation, but the absolute values are slightly shifted. e.g., $\text{BALD}[\mathbf{x}] \approx \text{BetaMarginalBALD}[\mathbf{x}] + err$ for some constant $err \in \mathbb{R}$. This study also implies that Beta marginal approximation is a reasonable assumption.

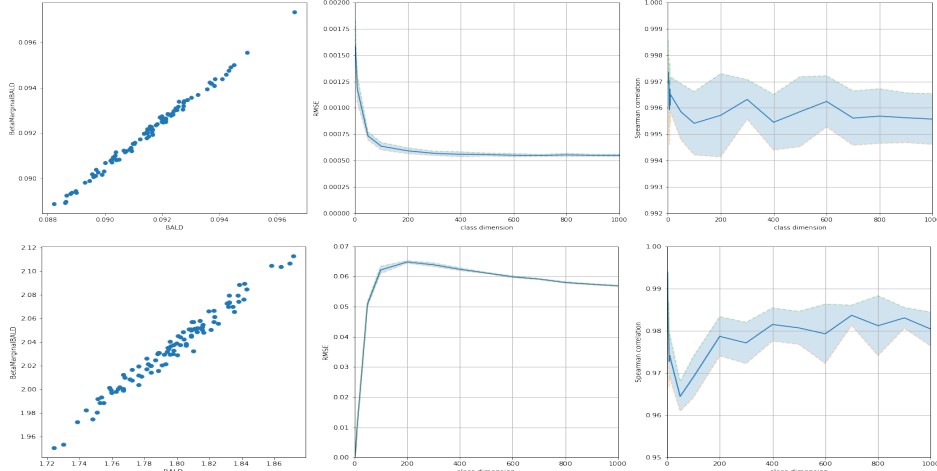

Figure 10: Scatter plot at $C = 1,000$ between BALD and BetaMarginalBALD (left), RMSE between BALD and BetaMarginalBALD over various class dimensions (middle), and Spearman's rank correlations over various class dimensions (right). The first row is the result from $C$-dimensional 100 random Dirichlet samples. The second row is obtained from softmax-applied 100 random Gaussian samples. Then we repeat the process 10 times. In both cases, we observe $> 96\%$ rank correlations as well.

## A.10 TOY EXAMPLE WITH EXPECTEDEFFECTIVELOSS AND BETAMARGINALEIG

As observed by high rank-correlation in Figure 8, BALD, ExpectedEffectiveLoss, and Beta-MarginalEIG show similar selections.

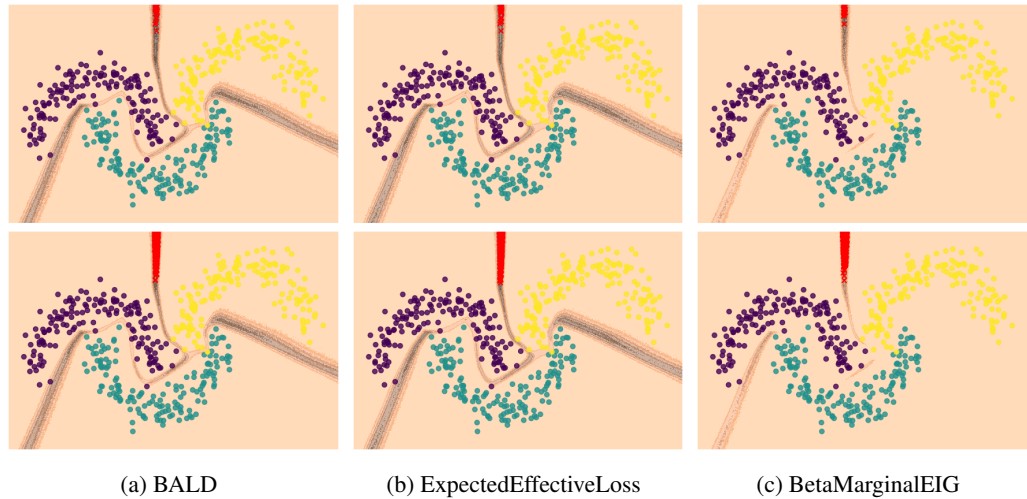

(a) BALD         (b) ExpectedEffectiveLoss         (c) BetaMarginalEIG

Figure 11: Top-$K$ selected points are marked by red color. The first row shows the top $K = 25$ point selections. The second row shows the top $K = 500$ point selections among around 0.6 million uniform grid points. The same experiments shown in Figure 2 in the main article.

Figure 12 shows the active learning curves for ExpectedEffectiveLoss and BetaMarginalEIG with MNIST and 3×CIFAR-100. This experiment also confirms that BALD and ExpectedEffectiveLoss are tightly aligned as we show that both are highly correlated. In MNIST, BetaMarginalEIG performs similar to BALD and ExpectedEffectiveLoss. However, in 3×CIFAR-100, BetaMarginalEIG performs similar to BALD at first, but it essentially performs better than BALD and similar to the random case. Recall that the rank correlation between BALD and BetaMarginalEIG is around 70%.

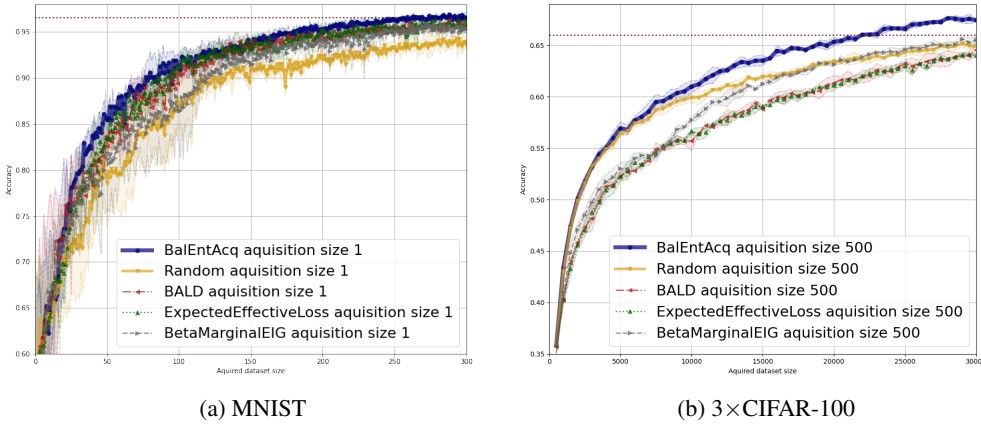

(a) MNIST

(b) 3×CIFAR-100

Figure 12: Active learning curves for ExpectedEffectiveLoss and BetaMarginalEIG with MNIST and 3×CIFAR-100.

### A.11 NON-NEGATIVE BALENTACQ REGION

In this section, we study the non-negative region of BalEntAcq $[\mathbf{x}]$. BalEntAcq$[\mathbf{x}]$ is non-negative when MJEnt$[\mathbf{x}] \geq 0$. Under Beta marginal distribution approximation, let $P_i \sim \text{Beta}(\alpha_i, \beta_i)$ in $\Phi(\mathbf{x}, \omega)$. Then we can fully write MJEnt$[\mathbf{x}]$ as follows:

$$\text{MJEnt}[\mathbf{x}] = \sum_i (\mathbb{E}P_i)\, h(P_i^+) + H(Y)$$

$$= \sum_{i=1}^{C} \left(\frac{\alpha_i}{\alpha_i + \beta_i}\right) \left[\log B(\alpha_i + 1, \beta_i) - \alpha_i \Psi(\alpha_i + 1) - (\beta_i - 1)\Psi(\beta_i) - (\alpha_i + \beta_i - 1)\Psi(\alpha_i + \beta_i + 1) - \log\left(\frac{\alpha_i}{\alpha_i + \beta_i}\right)\right].$$

Then, we are able to generate a 3D plot and a contour plot of BalEnt$[\mathbf{x}]$ when $C = 2$. i.e., $\Phi(\mathbf{x}, \omega) \sim \text{Dirichlet}(\alpha, \beta)$ such that $P_1 \sim \text{Beta}(\alpha, \beta)$ and $P_2 \sim \text{Beta}(\beta, \alpha)$.

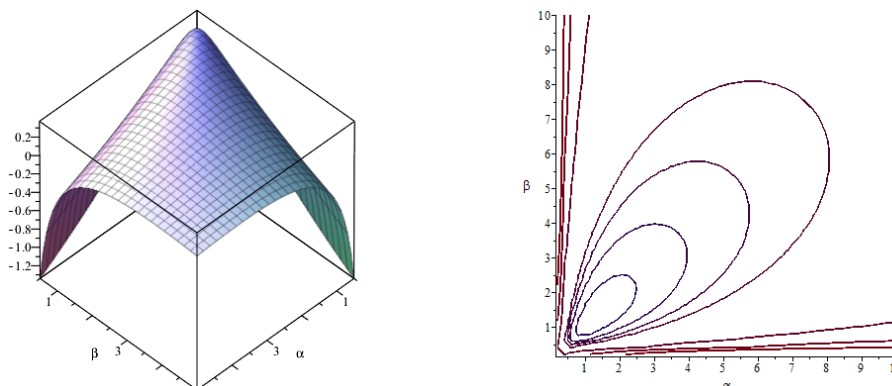

Figure 13: BalEnt 3D plot (left) and Positive BalEnt contour plot (right) over parameters $(\alpha, \beta)$. For the contour plot, starting from the outside, contours are generated when BalEnt$[\mathbf{x}] \in \{-3, -2, -1, 0, 0.1, 0.2, 0.3\}$.

Figure 13 suggests that in Dirichlet distribution's parameter space, there exist (uncountably and) infinitely many parameters which produce non-negative BalEnt$[\mathbf{x}]$ values. Then we also plot the non-negative region (red shaded) of BalEntAcq in our toy example.

Figure 14-(a) illustrates that there exist infinitely many points which produce the same BalEntAcq$[\mathbf{x}]$ values. Therefore we may imagine that we are conducting a uniform sampling on each contour surface $\{\text{BalEnt}[\mathbf{x}] = \lambda\}$ for each $\lambda \geq 0$, then moving to the surface for each $\lambda \geq 0$. This observation also explains how BalEnt$[\mathbf{x}]$ diversifies the selection near the decision boundary.

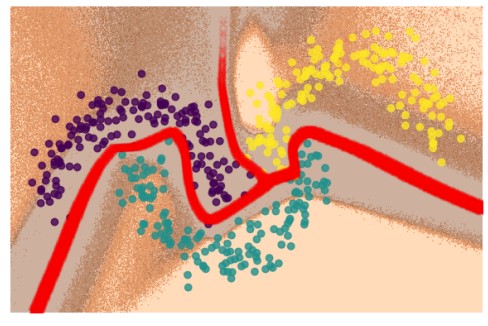 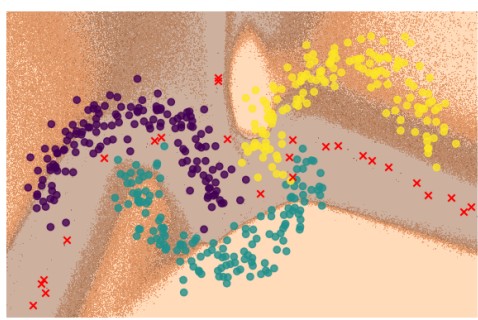

(a) Non-negative BalEntAcq region                    (b) Acquired 25 BalEntAcq points

Figure 14: Non-negative BalEntAcq region illustration from the toy example

## A.12 IMPLEMENTATION OF BALANCED ENTROPY ACTIVE LEARNING

Implementation of BalEntAcq active learning is almost the same as the usual MC dropout-based uncertainty methods. The difference is the dropout samples at inference time; BalEntAcq requires an additional estimation step of Beta parameters for each marginal distribution. Algorithm 1 explains the whole steps of BalEntAcq active learning. Moreover, we do not apply any early stopping criteria in each training because we understand that early stopping conflicts with dropout-based model training. In other words, if we stop too early in model training by observing validation accuracy or loss, we observe that model weights cannot reach to the fully mixing states of the randomness in MC dropouts, similar to the early stage of Markov Chain Monte Carlo (MCMC).

---

**Algorithm 1:** BalEntAcq active learning algorithm

---

1   **Input:** 1) Unlabeled dataset $\mathcal{D}_{\textbf{pool}}$, 2) initially labelled dataset $\mathcal{D}_{\textbf{training}}^{(0)}$, 3) the number of dropout samples $M$ at inference time, 4) active learning budget $K$ for each iteration, 5) total active learning budget $K^{tot}$

2   **Initialize** all weights of Bayesian neural network $\Phi$ and set $n \leftarrow 0$

3   **Repeat** at iteration $n \geq 0$

4       Train the model $\Phi$ with $\mathcal{D}_{\textbf{training}}^{(n)}$

5       For each $\mathbf{x} \in \mathcal{D}_{\textbf{pool}} \setminus \mathcal{D}_{\textbf{training}}^{(n)}$,

6          Generate $M$ dropout samples

7          Estimate Beta parameters $(\alpha_i, \beta_i)$ for each marginal distribution

8          Calculate BalEntAcq$[\mathbf{x}]$

9       Set $\mathcal{D}_{\textbf{training}}^{(n+1)} \leftarrow \mathcal{D}_{\textbf{training}}^{(n)} \bigcup \left\{ \text{top } K \text{ BalEntAcq-valued } \mathbf{x} \in \mathcal{D}_{\textbf{pool}} \setminus \mathcal{D}_{\textbf{training}}^{(n)} \right\}$, and $n \leftarrow n+1$

10  **Until** $\left| \mathcal{D}_{\textbf{training}}^{(n-1)} \right|$ reaches to $K^{tot}$

---

## A.13 MORE EXPERIMENTAL DETAILS

Table 2 shows a summary of dataset, configurations, and hyperparmeters used in our experiments. For each experiment, we repeat 3 times to generate the full active learning accuracy curve.

| Scenario | Dataset | # Classes | $K$ | $K^{tot}$ | Backbone | Loss | Image size | Batch size | Optimizer | Epochs | Learning rate | Dropout | MC trials |
|---|---|---|---|---|---|---|---|---|---|---|---|---|---|
| Full dropouts | MNIST | 10 | 1 | 300 | CNN | Cross-entropy | $28 \times 28$ | 128 | Adam | 150 | 0.01 | 50% | 100 |
| Fixed feature | CIFAR-100 | 100 | 500 | 10,000 | ResNet-50 | Cross-entropy | $224 \times 224$ | 128 | Adam | 150 | 0.0003 | 20% | 100 |
| Redundant images | CIFAR-100 | 100 | 500 | 30,000 | ResNet-50 | Cross-entropy | $224 \times 224$ | 128 | Adam | 150 | 0.0003 | 20% | 100 |
| Pre-trained backbone | TinyImageNet | 200 | 1,500 | 30,000 | ResNet-50 | Cross-entropy | $64 \times 64$ | 128 | Adam | 100 | 0.0003 | 20% | 100 |

Table 2: Detailed configurations used in our experiments.

In SimCLR (Chen et al., 2020a) feature training, we trained ResNet-50 with $224 \times 224$ image size, 192 batch size, 500 epochs, and 0.0003 learning rate with Adam optimizer for CIFAR-10/CIFAR-100.

### A.13.1 SIMPLY LAST+ LAYER BAYESIAN

Dropout-based Bayesian neural network typically requires adding dropout layer with ReLU activation for each convolutional or linear layer to approximate a Gaussian process (Gal & Ghahramani, 2016). But this requires a high computational cost. Therefore we adopt several additional last layer dropout architecture to build a Bayesian neural network equipped with Beta approximation. There exist several different lines of works to justify the effectiveness of this simple last layer modification (Snoek et al., 2015; Wilson et al., 2016; Brosse et al., 2020; Kristiadi et al., 2020; Hobbhahn et al., 2020). More precisely, similar to Laplace approximation applied at the last layer (Kristiadi et al., 2020, See Theorem 2.4) and (Hobbhahn et al., 2020), we may replace several last linear layers with a dropout applied and ReLU activated linear layers.

For example, we may add two or more dropout layers after ResNet-50 fixed backbone in our CIFAR-100 experiments to avoid any pathological cases (Foong et al., 2020). In practice, we observe a single dropout layer application is sufficient to achieve our Beta approximated marginals as shown below. We note that in MNIST experiment, we use $50\%$ dropout rate and for all other our experiments, we use $20\%$ dropout rate.

### A.13.2 CHOICES OF PRIORITIZATION IN BALENTACQ

In this section, we study the impact of the prioritization in $\text{BalEnt}[\mathbf{x}]$.

P1. $\text{P1}[\mathbf{x}] = -\text{BalEnt}[\mathbf{x}]$. This is the case where we put higher priority when the posterior uncertainty captures very small values. Note that this also includes high epistemic uncertainty (BALD) valued case. For example, when $C = 2$ with $\Phi(\mathbf{x}, \omega) \sim \text{Dirichlet}(\alpha, \beta)$, the posterior uncertainty goes to $-\infty$ as $\alpha \to 0$ and $\beta \to 0$. Therefore $\text{BalEnt}[\mathbf{x}] \to -\infty$. But this case also achieves the highest epistemic uncertainty.

P2. $\text{P2}[\mathbf{x}] = \begin{cases} \text{BalEnt}[\mathbf{x}]^{-1} & \text{if } \text{BalEnt}[\mathbf{x}] \geq 0, \\ \text{BalEnt}[\mathbf{x}] & \text{if } \text{BalEnt}[\mathbf{x}] < 0 \end{cases}$. This is the same case as our proposed acquisition measure.

P3. $\text{P3}[\mathbf{x}] = \text{BalEnt}[\mathbf{x}]$. This is the case where we put higher priority when the posterior uncertainty captures very high values (close to zero). As discussed in Section 4.1, we want to prioritize more when the information imbalance gap is higher.

Figure 15 and Table 3 show that selecting the points near $\text{BalEnt}[\mathbf{x}] \approx 0$ is a better way to improve the accuracy as we discussed in Section 4.1. When we prioritize the small posterior uncertainty case, P1 shows a very poor performance in a fixed feature scenario. However, under the backbone and augmentation scenario, the performance of P1 is similar to the high posterior uncertainty case of P3. This could be because of the evolution of the feature space and the batch normalization during the active learning process. i.e. previously captured uncertainty values will not be preserved under the backbone with augmentation scenario.

Table 3: Selected accuracy table depending on different prioritization. Mean and standard deviation are from 3 repeated experiments. The best performance in each column is shown in **bold**.

| Scenario | Redundant images + Fixed feature | | | | Backbone + Augmentation | | | |
|---|---|---|---|---|---|---|---|---|
| Dataset/Acq. Size/Test size | 3×CIFAR-100/500/10,000 | | | | TinyImageNet/1,500/10,000 | | | |
| Train Size/Pool Size | 5,000/150,000 | 15,000/150,000 | 25,000/150,000 | 30,000/150,000 | 6,000/100,000 | 15,000/100,000 | 24,000/100,000 | 30,000/100,000 |
| P1 | $37.6 \pm 0.7\%$ | $41.5 \pm 0.4\%$ | $45.6 \pm 1.1\%$ | $48.4 \pm 0.8\%$ | $28.2 \pm 1.2\%$ | $35.8 \pm 1.0\%$ | $41.7 \pm 0.7\%$ | $43.6 \pm 0.2\%$ |
| P2 (BalEntAcq) | $\mathbf{56.9 \pm 0.6\%}$ | $\mathbf{63.5 \pm 0.4\%}$ | $\mathbf{66.6 \pm 0.3\%}$ | $\mathbf{67.4 \pm 0.1\%}$ | $\mathbf{30.0 \pm 0.9\%}$ | $\mathbf{38.5 \pm 0.2\%}$ | $\mathbf{42.8 \pm 0.7\%}$ | $\mathbf{45.3 \pm 0.4\%}$ |
| P3 | $53.6 \pm 0.1\%$ | $60.5 \pm 0.6\%$ | $63.4 \pm 0.2\%$ | $64.7 \pm 0.2\%$ | $28.9 \pm 0.5\%$ | $36.2 \pm 0.9\%$ | $41.0 \pm 0.9\%$ | $43.6 \pm 0.2\%$ |

### A.13.3 BEHAVIOR OF DIFFERENT PRECISION LEVELS

As shown in the proof of Theorem 4.1, we may have some freedom to choose the level of the precision in the $P_i$ estimation. Therefore we report the active learning behavior for other precision choices. It is not clear which precision level achieves the optimal performance, but our preference

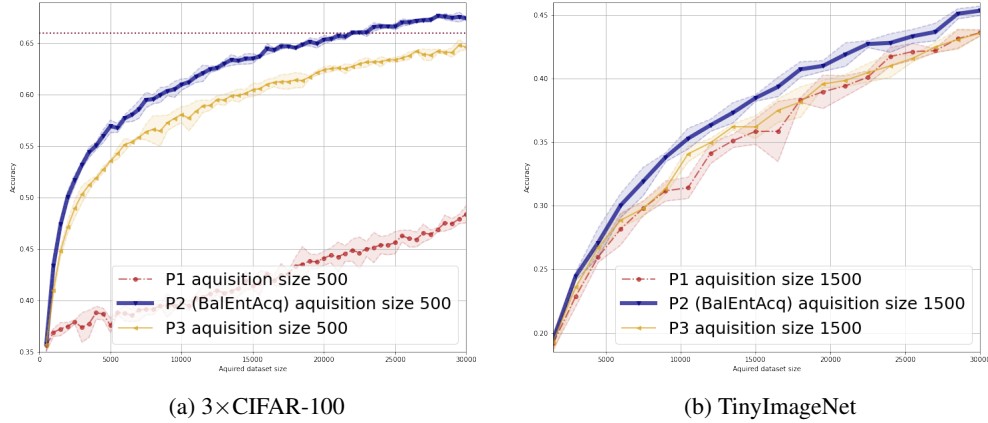

| (a) 3×CIFAR-100 | (b) TinyImageNet |

Figure 15: Active learning curves depending on different prioritization.

of $-\log \Delta \approx H(Y) + \log 2$ shows a reasonably superior performance in any scenario as shown in Figure 16 and Table 4.

Case 1. $-\log \Delta \approx H(Y) - \log 2$,

Case 2. $-\log \Delta \approx H(Y)$,

Case 3. $-\log \Delta \approx H(Y) + \log 2$ (our choice),

Case 4. $-\log \Delta \approx H(Y) + 2\log 2$,

Case 5. $-\log \Delta \approx H(Y) + 3\log 2$.

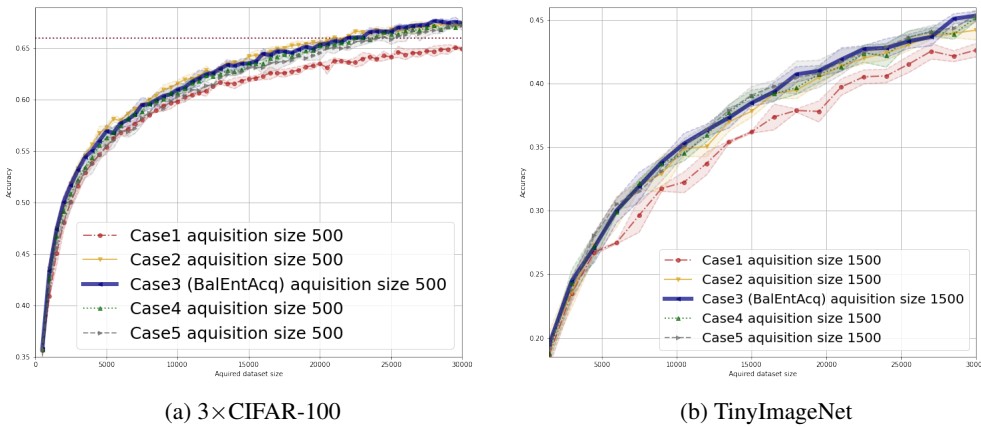

| (a) 3×CIFAR-100 | (b) TinyImageNet |

Figure 16: Active learning curves depending on different precision levels.

Table 4: Selected accuracy table depending on different precision levels. Mean and standard deviation are from 3 repeated experiments. The best performance in each column is shown in **bold**.

| Scenario | Redundant images + Fixed feature | | | | Backbone + Augmentation | | | |
|---|---|---|---|---|---|---|---|---|
| Dataset/Acq. Size/Test size | 3×CIFAR-100/500/10,000 | | | | TinyImageNet/1,500/10,000 | | | |
| Train Size/Pool Size | 5,000/150,000 | 15,000/150,000 | 25,000/150,000 | 30,000/150,000 | 6,000/100,000 | 15,000/100,000 | 24,000/100,000 | 30,000/100,000 |
| Case1 | $55.4 \pm 0.7\%$ | $62.0 \pm 0.3\%$ | $64.2 \pm 0.1\%$ | $65.0 \pm 0.1\%$ | $27.5 \pm 0.1\%$ | $36.2 \pm 0.1\%$ | $40.6 \pm 0.5\%$ | $42.6 \pm 0.5\%$ |
| Case2 | $\mathbf{57.2} \pm 0.3\%$ | $\mathbf{64.2} \pm 0.2\%$ | $\mathbf{66.9} \pm 0.3\%$ | $\mathbf{67.4} \pm 0.2\%$ | $29.9 \pm 1.2\%$ | $37.8 \pm 0.5\%$ | $42.4 \pm 0.5\%$ | $44.2 \pm 0.8\%$ |
| Case3 (BalEntAcq) | $56.9 \pm 0.6\%$ | $63.5 \pm 0.4\%$ | $66.6 \pm 0.3\%$ | $\mathbf{67.4} \pm 0.1\%$ | $30.0 \pm 0.9\%$ | $38.5 \pm 0.2\%$ | $42.8 \pm 0.7\%$ | $45.3 \pm 0.4\%$ |
| Case4 | $56.3 \pm 0.7\%$ | $63.6 \pm 0.2\%$ | $66.4 \pm 0.3\%$ | $\mathbf{67.4} \pm 0.2\%$ | $29.9 \pm 0.2\%$ | $39.0 \pm 0.5\%$ | $42.2 \pm 0.9\%$ | $45.2 \pm 0.3\%$ |
| Case5 | $55.6 \pm 0.6\%$ | $63.2 \pm 0.1\%$ | $65.9 \pm 0.5\%$ | $67.2 \pm 0.5\%$ | $\mathbf{30.5} \pm 0.5\%$ | $\mathbf{39.1} \pm 0.7\%$ | $\mathbf{42.9} \pm 0.3\%$ | $\mathbf{45.4} \pm 0.5\%$ |

### A.13.4 MORE DETAILS ABOUT THE MAIN EXPERIMENT

Figure 17 shows the full active learning curves, negative log-likelihood, average epistemic uncertainty for selected samples, and average aleatoric uncertainty for selected samples. We note that our proposed method selects neither high epistemic uncertainty (=model uncertainty) nor aleatoric uncertainty (=data uncertainty) samples. Nevertheless, BelEntAcq shows a good performance improvement during the active learning iterations. Furthermore, we observe that BelEntAcq keeps choosing low aleatoric uncertainty points but increasing epistemic uncertainty points.

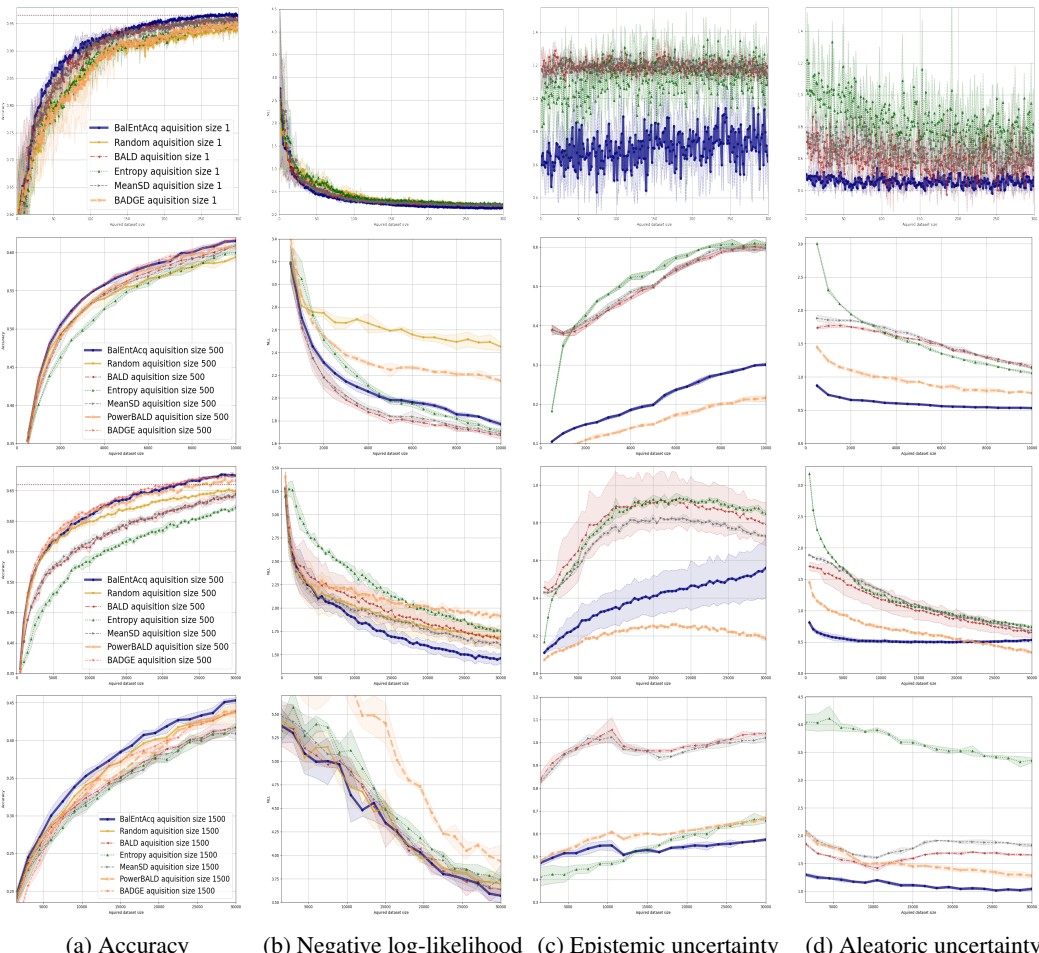

(a) Accuracy  (b) Negative log-likelihood  (c) Epistemic uncertainty  (d) Aleatoric uncertainty

Figure 17: Full active learning curves obtained from different scenarios. From the top row, each row represents a result of MNIST, CIFAR-100, 3×CIFAR-100, and TinyImageNet.

## A.14 TINYIMAGENET WITHOUT PRETRAINED MODEL

In this section, we report the active learning result of TinyImageNet without a pretrained model. By doing so, we can also observe the effect of the pre-knowledge of the model.

We use the same setting as we used in the main experiment. We observe that the accuracy progression is slower, so it requires more samples, but the observed behavior of each method is the same as before.

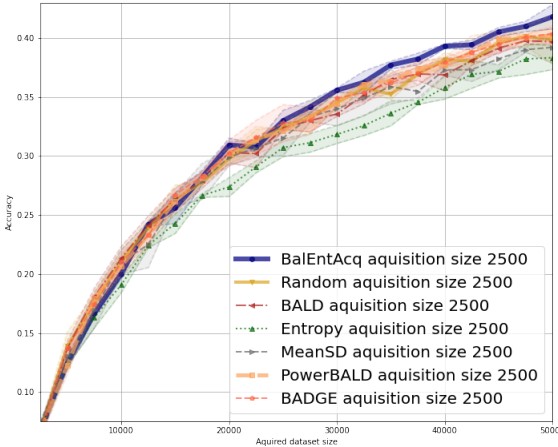

Figure 18: Active learning curves of TinyImageNet without pretrained model

Table 5: Selected accuracy table. Mean and standard deviation are from 3 repeated experiments.

| Scenario | Fixed feature + dropouts | | |
|---|---|---|---|
| Dataset/Acq. Size/Test size | TinyImageNet/2,500/100,000 | | |
| Train Size/Pool Size | 15,000/100,000 | 25,000/100,000 | 50,000/100,000 |
| Random | $\mathbf{26.3} \pm 1.2\%$ | $32.4 \pm 0.2\%$ | $39.9 \pm 0.2\%$ |
| BALD | $\mathbf{26.3} \pm 1.0\%$ | $32.4 \pm 0.2\%$ | $39.7 \pm 0.2\%$ |
| Entropy | $24.3 \pm 0.8\%$ | $30.7 \pm 0.8\%$ | $38.3 \pm 1.0\%$ |
| MeanSD | $\mathbf{26.3} \pm 0.5\%$ | $31.5 \pm 1.5\%$ | $39.2 \pm 1.3\%$ |
| PowerBALD | $26.2 \pm 0.5\%$ | $32.1 \pm 0.2\%$ | $40.2 \pm 0.8\%$ |
| BADGE (not-scalable) | $\mathbf{26.7} \pm 0.5\%$ | $\mathbf{32.8} \pm 1.6\%$ | $\mathbf{40.4} \pm 0.5\%$ |
| BalEntAcq (ours) | $25.6 \pm 1.3\%$ | $\mathbf{33.0} \pm 0.9\%$ | $\mathbf{41.8} \pm 1.0\%$ |

## A.15 RELATIONSHIP WITH THE EFFICIENT ACTIVE LEARNING ALGORITHM WITH ABSTENTION

This section illustrates the relationship with the recently proposed efficient Algorithm by Zhu & Nowak (2022b). Note that we are not proving the equivalence between the two algorithms. As demonstrated in A.11, we can see that our proposed BalEntAcq method shares a high similarity with active learning strategy with abstention (Locatelli et al., 2018; Shekhar et al., 2021; Puchkin & Zhivotovskiy, 2021; Zhu & Nowak, 2022a;b).

Intuitively, Algorithm 1 in Zhu & Nowak (2022b) works in the following way. Set an abstention parameter $\gamma > 0$. Train a binary classifier $h(x)$. For unlabelled point $\mathbf{x} \in \mathcal{X}$, we can calculate a uncertainty bound, $\text{UB}[\mathbf{x}] := [\text{lcb}(\mathbf{x}), \text{ucb}(\mathbf{x})]$. If $\text{UB}[\mathbf{x}] \subseteq \left[\frac{1}{2} - \gamma, \frac{1}{2} + \gamma\right]$, we abstain the point $\mathbf{x}$, i.e., we do not query the point $\mathbf{x}$. If $\frac{1}{2} \in \text{UB}[\mathbf{x}]$ and $\text{UB}[\mathbf{x}] \not\subseteq \left[\frac{1}{2} - \gamma, \frac{1}{2} + \gamma\right]$, we query the point $\mathbf{x}$. At each iteration $m$, we add geometrically increasing $2^m$ queried points.

The key insight of this Algorithm 1 to achieve exponential label savings is to abstain from the point very close to the decision boundary. Similarly, as we demonstrated in A.11, our BalEntAcq[$\mathbf{x}$] finds

a margin by focusing on the positive sign of BalEntAcq[$\mathbf{x}$] which corresponds to finding $\mathbf{x}$ outside the abstention region such that $\left| x - \frac{1}{2} \right| > \gamma$ near the decision boundary. Then following the positive BalEntAcq[$\mathbf{x}$] values, we acquire points toward the decision boundary direction, which corresponds to the condition $\frac{1}{2} \in$ UB[$\mathbf{x}$]. We know that the point near the decision boundary should have high aleatoric uncertainty (so possibly noise-seeking). On the other hand, Corollary 1 implies that aleatoric uncertainty is increasing as $\alpha, \beta \to +\infty$. So MJEnt[$\mathbf{x}$] $\to -\infty$. Then BalEntAcq[$\mathbf{x}$] $\to -\infty$. Therefore, our BalEntAcq[$\mathbf{x}$] will acquire points near the decision boundary but will not acquire the point if it's too close to the decision boundary. This strategy in our BalEntAcq[$\mathbf{x}$] exactly matches the key insight of Algorithm 1. So we may be able to theoretically guarantee that our proposed acquisition function BalEntAcq[$\mathbf{x}$] could be a universally working active learning algorithm by achieving exponential label savings.

## A.16 MORE EXPERIMENTS WITH SMALLER ACQUISITION SIZE

In this section, we conduct more experiments with $3\times$MNIST and $3\times$CIFAR-10 by adding more baselines such as VarRatio [$\mathbf{x}$] $:= 1 - \max_i \mathbb{E}P_i$ (Freeman, 1965), BatchBALD, and CoreSet. The main purpose of these experiments is to test the relatively smaller acquisition size. We acquire 25 points for each active learning iteration. For $3\times$MNIST, we use CNN architectures. For $3\times$CIFAR-10, we fix the feature space obtained from SimCLR (Chen et al., 2020a), the same setting we used in our main experiments. Overall, the additional experimental results are well-aligned with our main results. We observe that BADGE is the best performing baseline. However, we note that BADGE is not linearly scalable, and it requires more computational costs. Figure 19 and Table 6 show full results.

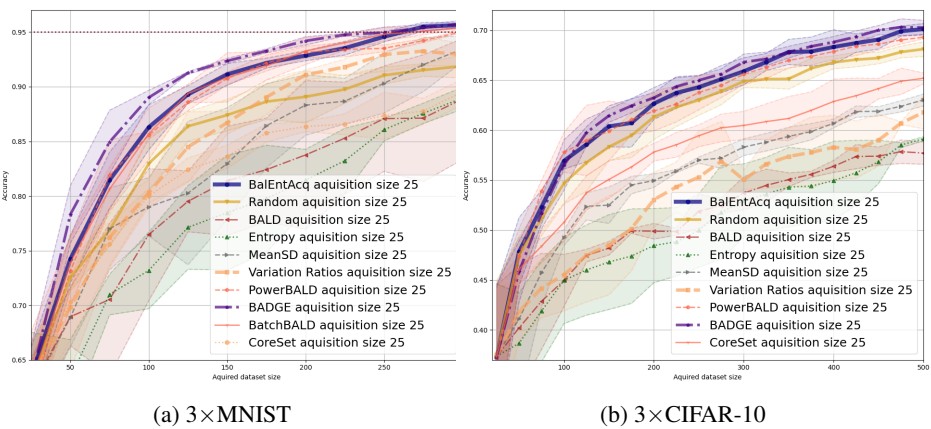

(a) $3\times$MNIST  (b) $3\times$CIFAR-10

Figure 19: Active learning curves of smaller batch size with $3\times$MNIST and $3\times$CIFAR-10.

Table 6: Selected accuracy table. Mean and standard deviation are from 3 repeated experiments.

| Scenario | Full dropouts + CNN | | | Fixed feature | | |
|---|---|---|---|---|---|---|
| Dataset/Acq. Size/Test size | $3\times$MNIST/25/10,000 | | | $3\times$CIFAR-10/25/10,000 | | |
| Train Size/Pool Size | 100/180,000 | 200/180,000 | 300/180,000 | 200/150,000 | 300/150,000 | 500/150,000 |
| Random | $83.0 \pm 2.5\%$ | $89.1 \pm 2.4\%$ | $91.9 \pm 1.1\%$ | $61.3 \pm 2.3\%$ | $64.9 \pm 0.6\%$ | $68.1 \pm 0.7\%$ |
| BALD | $76.5 \pm 6.0\%$ | $83.8 \pm 2.8\%$ | $88.9 \pm 5.5\%$ | $49.8 \pm 0.8\%$ | $53.7 \pm 1.7\%$ | $57.7 \pm 1.1\%$ |
| Entropy | $73.2 \pm 3.5\%$ | $81.5 \pm 2.8\%$ | $89.0 \pm 1.0\%$ | $48.5 \pm 3.7\%$ | $53.0 \pm 3.7\%$ | $59.1 \pm 0.7\%$ |
| MeanSD | $79.1 \pm 2.7\%$ | $88.3 \pm 2.8\%$ | $93.5 \pm 1.2\%$ | $55.0 \pm 0.7\%$ | $58.3 \pm 0.8\%$ | $63.0 \pm 0.6\%$ |
| Variation Ratios | $80.3 \pm 1.3\%$ | $91.1 \pm 0.5\%$ | $92.9 \pm 1.0\%$ | $53.0 \pm 2.7\%$ | $55.0 \pm 1.0\%$ | $61.8 \pm 1.6\%$ |
| PowerBALD | $85.6 \pm 0.4\%$ | $93.2 \pm 0.5\%$ | $95.0 \pm 0.1\%$ | $63.2 \pm 0.5\%$ | $66.8 \pm 0.8\%$ | $69.3 \pm 0.2\%$ |
| BADGE (not-scalable) | $\mathbf{89.0} \pm 0.7\%$ | $\mathbf{94.2} \pm 0.4\%$ | $\mathbf{95.8} \pm 0.3\%$ | $\mathbf{63.2} \pm 0.5\%$ | $\mathbf{66.8} \pm 0.8\%$ | $\mathbf{70.3} \pm 0.7\%$ |
| BatchBALD (not-scalable) | $85.7 \pm 2.4\%$ | $93.3 \pm 1.2\%$ | $95.4 \pm 0.2\%$ | $-$ | $-$ | $-$ |
| CoreSet (not-scalable) | $80.0 \pm 0.7\%$ | $86.4 \pm 0.5\%$ | $89.5 \pm 1.2\%$ | $57.8 \pm 0.9\%$ | $60.5 \pm 1.4\%$ | $65.2 \pm 0.5\%$ |
| BalEntAcq (ours) | $\mathbf{86.3} \pm 2.0\%$ | $\mathbf{92.8} \pm 0.5\%$ | $\mathbf{95.7} \pm 0.2\%$ | $\mathbf{62.7} \pm 1.6\%$ | $\mathbf{65.9} \pm 1.4\%$ | $\mathbf{70.1} \pm 0.5\%$ |

### A.17 COMPARISON WITH COREMSE

Recently, the Bayesian active learning framework considering the Expected Loss Reduction (ELR) for the optimal Bayes classifier has been proposed (Zhao et al., 2021; Tan et al., 2021). Under this framework, they attempt to optimize the loss reduction in a holistical way, accounting for average loss reduction from all points. However, this non-parametric approach requires a very expensive computational cost. With a large dataset size, ELR (Zhao et al., 2021), wMOCU (Zhao et al., 2021), CoreLog (Tan et al., 2021), and CoreMSE (Tan et al., 2021) require a vast memory size unless we apply size reductions on the data space and MC samples (Tan et al., 2021). If the number of classes is large, running the algorithm in practice is impossible. Therefore the naive application of the ELR-based algorithm is not scalable. Moreover, both works have pitfalls in the convergence proof by assuming the finite data and parameter space. Both pieces of the works end up with null proof. Nevertheless, we tested the performance of CoreMSE (Tan et al., 2021) with MNIST, seemingly the best method under this framework. Figure 20 and Table 7 show the full active learning results.

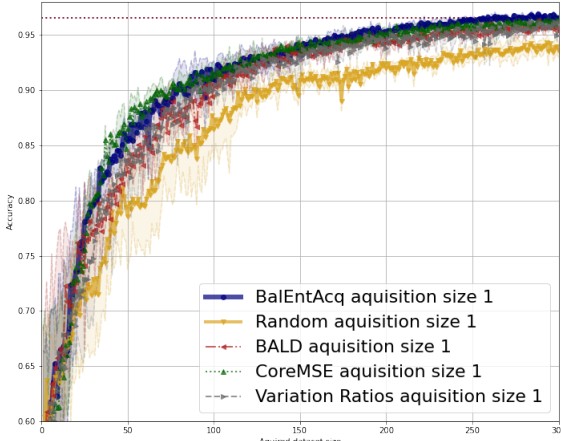

Figure 20: Active learning curves with CoreMSE in MNIST

Table 7: Selected accuracy table. Mean and standard deviation are from 3 repeated experiments.

| Scenario | Full dropouts + CNN | | |
|---|---|---|---|
| Dataset/Acq. Size/Test size | MNIST/1/10,000 | | |
| Train Size/Pool Size | 50/60,000 | 100/60,000 | 300/60,000 |
| Random | $78.6 \pm 4.9\%$ | $86.4 \pm 2.7\%$ | $93.6 \pm 0.7\%$ |
| BALD | $82.6 \pm 1.3\%$ | $90.5 \pm 0.8\%$ | $95.3 \pm 0.4\%$ |
| Variation Ratios | $83.4 \pm 3.2\%$ | $90.0 \pm 1.2\%$ | $96.2 \pm 0.2\%$ |
| CoreMSE | $\mathbf{86.5} \pm 0.8\%$ | $91.3 \pm 0.3\%$ | $96.4 \pm 0.3\%$ |
| BalEntAcq (ours) | $85.4 \pm 1.0\%$ | $\mathbf{91.4} \pm 1.3\%$ | $\mathbf{96.5} \pm 0.1\%$ |

### A.18 ACQUISITION TIME AND SPACE COMPLEXITY

In this section, we discuss the time and space complexity of the acquisition calculation for each active learning iteration. We denote by $N$ number of unlabelled points, $C$ number of classes, $K$ the acquisition size. For BADGE, we use the last layer feature vector and then apply k-means++ initialization (Vassilvitskii & Arthur, 2007).

The main computational bottleneck of BADGE is the dependency of $K$ in the calculation of multi-acquisition. Here, the $K$-iteration cannot be parallelized since the pairwise distance calculation depends on the previous selection of centers in k-means++. On the other hand, the main computational bottleneck of CoreMSE is the dependency of $N^2$ in both time and space complexity. Since the

Table 8: First two rows show the theoretical time and space complexity. Remaining rows present the average calculation time what we observed in our experiments.

| Method | BalEntAcq | BALD | Entropy | MeanSD | PowerBALD | BADGE | CoreMSE | Random |
|---|---|---|---|---|---|---|---|---|
| Time Complexity | $O(CN)$ | $O(CN)$ | $O(CN)$ | $O(CN)$ | $O(CN)$ | $O(CNK)$ | $O(C^2N^2K)$ | $O(N)$ |
| Space Complexity | $O(C+K)$ | $O(C+K)$ | $O(C+K)$ | $O(C+K)$ | $O(C+K)$ | $O(C+N)$ | $O(C^2N^2)$ | $O(1)$ |
| Case | | | | Average Elapsed Time (sec) | | | | |
| MNIST with Acq. size 1 | 7.1 | 6.9 | 6.8 | 6.3 | – | – | 14.3 | 0.1 |
| CIFAR-100 with Acq. size 500 | 10.2 | 9.4 | 9.5 | 9.6 | 9.6 | 302.5 | insufficient memory | 0.1 |
| 3×CIFAR-100 with Acq. size 500 | 18.9 | 18.5 | 18.4 | 18.2 | 18.4 | 1227.4 | insufficient memory | 0.3 |
| SVHN with Acq. size 2500 | 20.7 | 19.7 | 19.3 | 19.4 | 20.3 | 85.4 | insufficient memory | 0.1 |
| TinyImageNet with Acq. size 1500 | 183.4 | 178.7 | 178.4 | 176.4 | 182.0 | 4936.1 | insufficient memory | 0.2 |

original CoreMSE requires calculating the entire correlational effect on the loss from all data points, we cannot linearly scale up this computation. Even if we try to relax the memory issue by fixing the number of points of interest for correlation instead of the entire unlabelled dataset, we still experience a memory deficiency.

The following Figure 21 confirms the dependency of the acquisition time of our BalEntAcq and BADGE concerning the unlabelled dataset size and the acquisition size by augmenting the number of images from CIFAR-10 and CIFAR-100. Our BalEntAcq does not rely on the acquisition size, but BADGE is still linearly proportional to the acquisition size. For the unlabelled dataset size, both methods linearly increase the acquisition time, but the slope of BADGE is much steep.

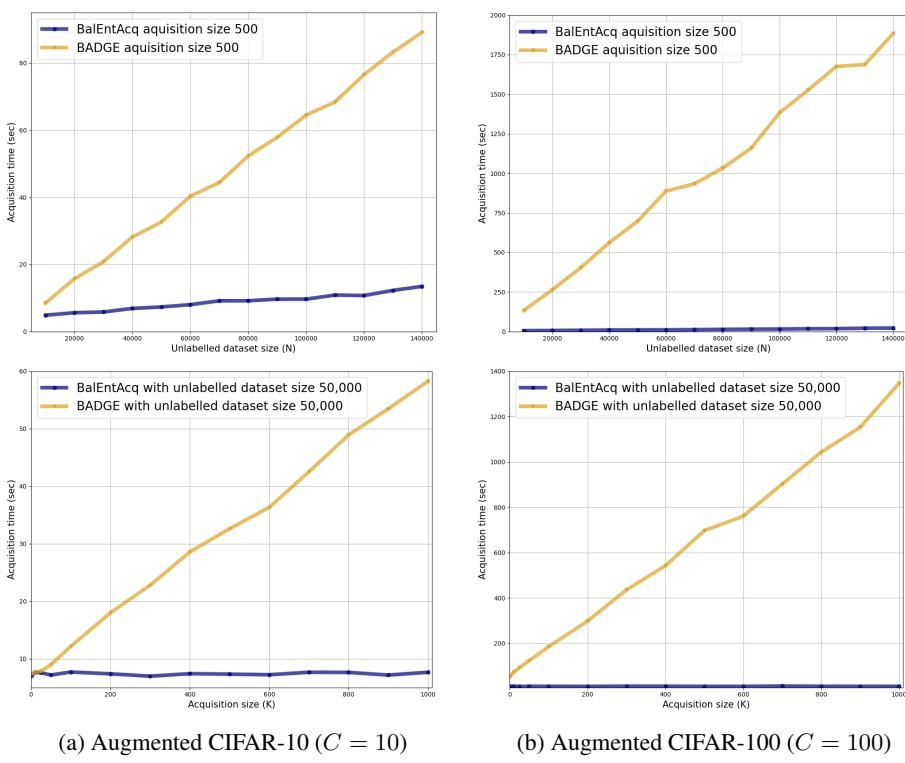

(a) Augmented CIFAR-10 ($C = 10$)  (b) Augmented CIFAR-100 ($C = 100$)

Figure 21: Acquisition time under different unlabelled dataset sizes (first row) and under different acquisition sizes (second row).

## A.19 EXPERIMENTS UNDER HEAVILY REDUNDANT DATA SCENARIO

In this section, we observe the active learning performance behavior to various redundancy levels with $\tau \times$CIFAR-100 dataset where $\tau \in \{1, 3, 5, 7, 10, 20, 50\}$.

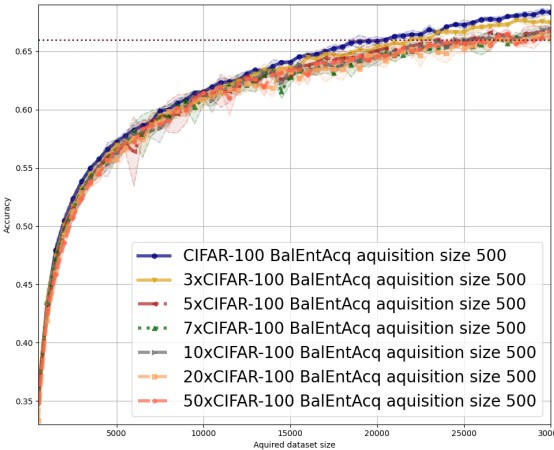

Figure 22: Active learning curves under various redundancy level scenarios

Active learning performance with BalEntAcq shows the best accuracy when no redundant images exist. With $50\times$CIFAR-100, our BalEntAcq requires only up to $1.25\%$ data points among 2.5 million images to achieve the fully supervised accuracy ($= 66\%$). We do not observe any significant performance deterioration compared to the increment of the pool size, even in highly redundant image scenarios. It demonstrates the possibility of exponential label savings as discussed in Section A.15. Refer to the following Table 9 for the specific accuracy values.

Table 9: Selected accuracy table for various redundancy levels. Mean and standard deviation are from 3 repeated experiments.

| Scenario | Fixed feature + heavily redundant data scenario | | | | | |
|---|---|---|---|---|---|---|
| Acq. Size/Test size | 500/10,000 | | | | | |
| Train Size | 5,000 | 10,000 | 15,000 | 20,000 | 25,000 | 30,000 |
| CIFAR-100/50,000 | $\mathbf{57.2} \pm 0.2\%$ | $\mathbf{61.5} \pm 0.2\%$ | $\mathbf{64.1} \pm 0.1\%$ | $\mathbf{65.9} \pm 0.4\%$ | $\mathbf{67.3} \pm 0.4\%$ | $\mathbf{68.3} \pm 0.2\%$ |
| $3\times$CIFAR-100/150,000 | $56.9 \pm 0.6\%$ | $61.0 \pm 0.5\%$ | $63.5 \pm 0.5\%$ | $65.4 \pm 0.4\%$ | $66.7 \pm 0.3\%$ | $67.4 \pm 0.1\%$ |
| $5\times$CIFAR-100/250,000 | $56.3 \pm 0.9\%$ | $61.4 \pm 0.3\%$ | $62.8 \pm 0.1\%$ | $65.1 \pm 0.6\%$ | $65.6 \pm 0.1\%$ | $66.8 \pm 0.4\%$ |
| $7\times$CIFAR-100/350,000 | $56.6 \pm 0.9\%$ | $61.0 \pm 0.5\%$ | $62.8 \pm 0.6\%$ | $64.6 \pm 0.5\%$ | $65.9 \pm 0.3\%$ | $66.2 \pm 0.4\%$ |
| $10\times$CIFAR-100/500,000 | $56.3 \pm 0.4\%$ | $61.3 \pm 0.2\%$ | $63.4 \pm 0.5\%$ | $64.4 \pm 0.7\%$ | $66.0 \pm 0.4\%$ | $66.9 \pm 0.2\%$ |
| $20\times$CIFAR-100/1,000,000 | $55.7 \pm 0.8\%$ | $60.9 \pm 0.7\%$ | $63.2 \pm 0.5\%$ | $64.6 \pm 0.8\%$ | $65.3 \pm 0.4\%$ | $66.1 \pm 0.6\%$ |
| $50\times$CIFAR-100/2,500,000 | $56.5 \pm 0.7\%$ | $61.4 \pm 0.3\%$ | $63.4 \pm 0.5\%$ | $64.6 \pm 0.2\%$ | $65.7 \pm 0.5\%$ | $66.8 \pm 0.2\%$ |

## A.20 DISCUSSION ABOUT BALENTACQ FORM

In this section, we discuss further the underlying motivation of the ratio form of our BalEnt[$\mathbf{x}$]:

$$\text{BalEnt}[\mathbf{x}] := \frac{\text{MJEnt}[\mathbf{x}]}{\text{Ent}[\mathbf{x}] + \log 2} = \frac{\sum_i \left(\mathbb{E}P_i\right) h(P_i^+) + H(Y)}{H(Y) + \log 2}.$$

We design BalEnt[$\mathbf{x}$] to be a generalized probability as a ratio between the marginalized joint entropy MJEnt[$\mathbf{x}$] and the augmented Shannon entropy $H(Y)+\log 2$. As shown in Appendix A.5 for the proof of Theorem 4.1, BalEnt[$\mathbf{x}$] has a natural interpretation as a generalized estimation error probability leveraging Fano's inequality. i.e., $-\infty < \text{BalEnt}[\mathbf{x}] \leq 1$. The rigorous interpretation of the negative probability is an open problem. Still, we can intuitively understand that the negative probability implies that the model has sufficient knowledge to estimate the value of $P_i^+$ given the information of $Y$ up to the specified precision level. It suggests that a phase transition in the model's current knowledge occurs near the zero probability point of BalEnt[$\mathbf{x}$]. It is because $H(Y)$ is the minimum amount of nats (or bits) to encode the information of $Y$, which is the fundamental information limit when observing labels, and we require an additional $\log 2$ nats given our choice of the precision level. That's why we call the quantity BalEnt[$\mathbf{x}$] to be a balanced entropy that captures the information balance between the model and the label.

Moreover, although we empirically demonstrate the choice of the precision level $H(Y) + \log 2$ to match MJEnt[$\mathbf{x}$] in Appendix A.13.3, MJEnt[$\mathbf{x}$] itself is a result of the derivation by applying Jensen's inequality shown in Appendix A.2. In other words, MJEnt[$\mathbf{x}$] is the maximally achievable joint entropy between $\Phi(\mathbf{x}, \omega)$ and $Y(\mathbf{x}, \omega)$. So we may expect that there should exist an entropy-increasing physical dynamical process, something similar to a diffusion process (Oksendal, 2013), from the state $(\Phi(\mathbf{x}, \omega), Y(\mathbf{x}, \omega))$ to another maximally marginalized state $S'$ which has a joint entropy MJEnt[$\mathbf{x}$]. Because of our limited understanding of the point process entropy, we have yet to know the full specification of the existence of these physical dynamics.

Lastly, because our generalized probability interpretation of BalEnt[$\mathbf{x}$] mainly relies on the sign of BalEnt[$\mathbf{x}$], one might wonder if the denominator $H(Y) + \log 2$ of BalEnt[$\mathbf{x}$] is a critical factor. So we empirically demonstrate the necessity of the denominator $H(Y) + \log 2$.

We define an acquisition function MJEntAcq[$\mathbf{x}$] from MJEnt[$\mathbf{x}$] without the denominator of BalEnt[$\mathbf{x}$]:

$$
\text{MJEntAcq}[\mathbf{x}] := \begin{cases} \text{MJEnt}[\mathbf{x}]^{-1} & \text{if MJEnt}[\mathbf{x}] \geq 0, \\ \text{MJEnt}[\mathbf{x}] & \text{if MJEnt}[\mathbf{x}] < 0, \end{cases}
$$

Figure 23 and Table 10 show the active learning performances on CIFAR-100 and 3×CIFAR-100. We observe that MJEntAcq[$\mathbf{x}$] performs well to reach the fully supervised accuracy. However, BalEntAcq[$\mathbf{x}$] performs better than MJEntAcq[$\mathbf{x}$]. It suggests that the slight distortion of MJEnt[$\mathbf{x}$] to BalEnt[$\mathbf{x}$] could help improve active learning performance.

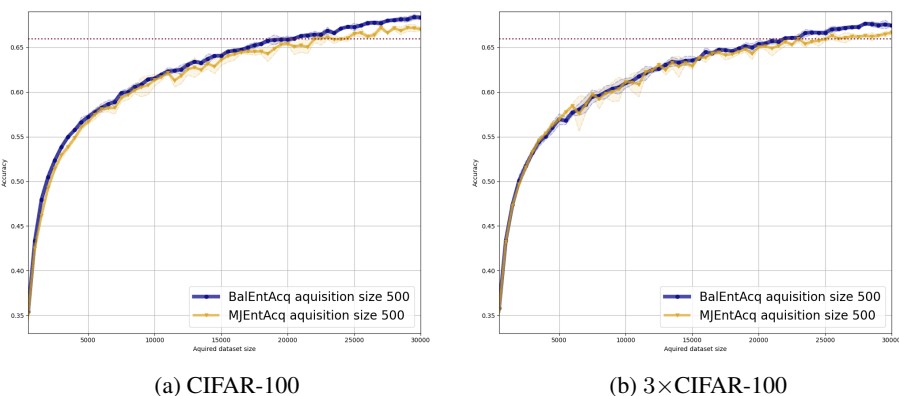

(a) CIFAR-100  (b) 3×CIFAR-100

Figure 23: Active learning curve comparison without denominator of BalEnt[$\mathbf{x}$]

Table 10: Selected accuracy table for various redundancy levels. Mean and standard deviation are from 3 repeated experiments.

| Scenario | Fixed feature + different acquisition form | | | | | |
|---|---|---|---|---|---|---|
| Dataset/Acq. Size/Test size | CIFAR-100/500/10,000 | | | | | |
| Train Size/Pool Size | 5,000/50,000 | 10,000/50,000 | 15,000/50,000 | 20,000/50,000 | 25,000/50,000 | 30,000/50,000 |
| BalEntAcq | **57.2** ± 0.2% | **61.5** ± 0.2% | **64.1** ± 0.1% | **65.9** ± 0.4% | **67.3** ± 0.4% | **68.3** ± 0.2% |
| MJEntAcq | 56.6 ± 0.2% | 61.4 ± 0.6% | 63.6 ± 0.6% | 65.4 ± 0.2% | 66.5 ± 0.2% | 67.0 ± 0.2% |
| Dataset/Acq. Size/Test size | 3×CIFAR-100/500/10,000 | | | | | |
| Train Size/Pool Size | 5,000/150,000 | 10,000/150,000 | 15,000/150,000 | 20,000/150,000 | 25,000/150,000 | 30,000/150,000 |
| BalEntAcq | **56.9** ± 0.6% | 61.0 ± 0.5% | **63.5** ± 0.5% | **65.4** ± 0.4% | **66.7** ± 0.3% | **67.4** ± 0.1% |
| MJEntAcq | **56.9** ± 0.3% | **61.1** ± 0.2% | 62.9 ± 0.4% | 65.0 ± 0.1% | 65.9 ± 0.2% | 66.7 ± 0.2% |

### A.21 Application to Bayesian neural network with variational dropouts

In this section, we report our active learning experiment when we train a Bayesian neural network with variational dropouts (Kingma et al., 2015) with 3×CIFAR-10 with acquisition size 50 and 3×CIFAR-100 with acquisition size 500 under a fixed feature scenario.

We use an Adam optimizer with a learning rate of 0.0003 and 500 epochs in each experiment. Compared to MC-dropout Bayesian neural network models, we observe that the convergence with variational dropouts is not stable, so it requires much longer epochs if we newly train the model at each active learning iteration for both cases. Therefore, we continue to train the model from the previously trained model at each iteration except the initial iteration so that the convergence can be more stable.

Here is the architecture we used for our 3×CIFAR-10 experiment, and for the 3×CIFAR-10 case, we can simply modify the out feature size to be 100.

```
VARIATIONAL_DROPOUT_CLASSIFIER(
  (classifier): Sequential(
    (0): VariationalDropout(in_features=2048, out_features=1024)
    (1): VariationalDropout(in_features=1024, out_features=1024)
    (2): Linear(in_features=1024, out_features=10, bias=False)
  )
)
```

We observe a similar result from the MC-dropout Bayesian neural networks. Our BalEntAcq consistently outperforms other linearly scalable baselines and is eventually on par with BADGE in 3×CIFAR-10 and approaching close to BADGE in 3×CIFAR-100.

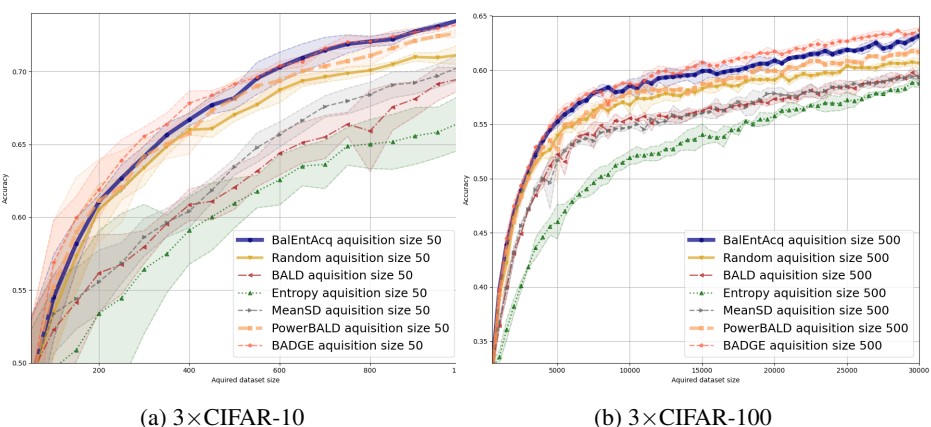

(a) 3×CIFAR-10                    (b) 3×CIFAR-100

Figure 24: Active learning curves with variational dropouts

Table 11: Selected accuracy table. Mean and standard deviation are from 3 repeated experiments.

| Scenario | Fixed feature + variational dropouts | | | | | |
|---|---|---|---|---|---|---|
| Dataset/Acq. Size/Test size | 3×CIFAR-10/50/10,000 | | | 3×CIFAR-100/500/10,000 | | |
| Train Size/Pool Size | 500/150,000 | 750/150,000 | 1000/150,000 | 5000/150,000 | 15,000/150,000 | 30,000/150,000 |
| Random | $67.0 \pm 0.5\%$ | $69.9 \pm 0.4\%$ | $71.2 \pm 0.7\%$ | $53.9 \pm 0.5\%$ | $58.3 \pm 0.2\%$ | $60.6 \pm 0.5\%$ |
| BALD | $62.1 \pm 1.5\%$ | $66.4 \pm 0.8\%$ | $69.5 \pm 0.8\%$ | $52.3 \pm 0.1\%$ | $56.3 \pm 0.2\%$ | $59.3 \pm 0.3\%$ |
| Entropy | $60.8 \pm 1.7\%$ | $64.9 \pm 1.3\%$ | $66.5 \pm 1.9\%$ | $46.0 \pm 1.6\%$ | $54.1 \pm 1.0\%$ | $58.8 \pm 0.4\%$ |
| MeanSD | $63.5 \pm 0.6\%$ | $68.0 \pm 0.9\%$ | $70.3 \pm 0.9\%$ | $51.6 \pm 0.2\%$ | $56.0 \pm 0.3\%$ | $59.4 \pm 0.7\%$ |
| PowerBALD | $68.2 \pm 1.3\%$ | $70.7 \pm 0.4\%$ | $72.6 \pm 0.3\%$ | $52.7 \pm 0.3\%$ | $58.7 \pm 0.1\%$ | $61.7 \pm 0.4\%$ |
| BADGE (not-scalable) | $\mathbf{69.1} \pm 0.1\%$ | $\mathbf{72.0} \pm 0.4\%$ | $73.2 \pm 0.1\%$ | $\mathbf{55.7} \pm 0.5\%$ | $\mathbf{60.7} \pm 0.1\%$ | $\mathbf{63.7} \pm 0.3\%$ |
| BalEntAcq (ours) | $\mathbf{68.2} \pm 0.8\%$ | $\mathbf{71.9} \pm 0.6\%$ | $\mathbf{73.5} \pm 0.2\%$ | $\mathbf{55.2} \pm 0.5\%$ | $59.9 \pm 0.9\%$ | $\mathbf{63.2} \pm 0.2\%$ |

## A.22 APPLICATION TO GAUSSIAN PROCESS

In this section, we report the active learning result with 3×CIFAR-10 when we train the model with the exact Gaussian process which could be of another independent interest.

We use an Adam optimizer with a learning rate of 0.1 and 5,000 iterations in each experiment. We use the exact Gaussian process. We observe that the exact Gaussian process requires much more

samples compared to the Bayesian neural network, but the observed behavior of each method is the same as before.

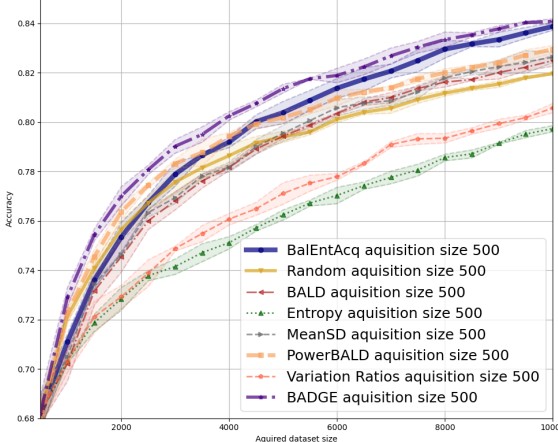

Figure 25: Active learning curves with exact Gaussian process

Table 12: Selected accuracy table. Mean and standard deviation are from 3 repeated experiments.

| Scenario | Fixed feature + variational dropouts | | |
|---|---|---|---|
| Dataset/Acq. Size/Test size | 3×CIFAR-10/500/10,000 | | |
| Train Size/Pool Size | 5,000/150,000 | 7,500/150,000 | 10,000/150,000 |
| Random | $79.4 \pm 0.2\%$ | $80.9 \pm 0.1\%$ | $82.0 \pm 0.3\%$ |
| BALD | $79.4 \pm 0.1\%$ | $81.4 \pm 0.1\%$ | $82.5 \pm 0.2\%$ |
| Entropy | $76.2 \pm 0.2\%$ | $78.1 \pm 0.2\%$ | $79.7 \pm 0.1\%$ |
| MeanSD | $79.6 \pm 0.3\%$ | $81.2 \pm 0.3\%$ | $82.6 \pm 0.2\%$ |
| Variation Ratio | $77.1 \pm 0.4\%$ | $79.3 \pm 0.2\%$ | $80.6 \pm 0.2\%$ |
| PowerBALD | $80.2 \pm 0.2\%$ | $81.8 \pm 0.2\%$ | $82.9 \pm 0.2\%$ |
| BADGE (not-scalable) | $\mathbf{81.4} \pm 0.2\%$ | $\mathbf{83.0} \pm 0.2\%$ | $\mathbf{84.1} \pm 0.1\%$ |
| BalEntAcq (ours) | $\mathbf{80.4} \pm 0.5\%$ | $\mathbf{82.5} \pm 0.5\%$ | $\mathbf{83.9} \pm 0.2\%$ |

### A.23 WEAK SIMILARITY WITH RISK PARITY PORTFOLIO OPTIMIZATION

This section is moderately informal and a little digression, but we think it is worth discussing. An active learning problem shares a weak similarity with a portfolio optimization problem (Markowitz, 1952). In portfolio optimization, we aim to minimize uncertainty but maximize profits under correlated environments. Traditional mean-variance portfolio optimization is well-known, and the primary objective function is minimizing the uncertainty (=risk), but the selection needs to be well-diversified (Markowitz, 1952). In active learning, we typically aim to maximize the uncertainty, but diversify the selection under correlated data points, e.g., BADGE (Ash et al., 2020).

In the modern dynamic financial system, the traditional way of optimization could still be exposed to a concentration risk (Levy & Zhang, 2019). To mitigate the concentration risk, equalizing the risk contribution, a.k.a. risk parity strategy, for each factor has been popularized after the financial crisis (Prince, 2011; Hurst et al., 2010; Chaves et al., 2011; Qian, 2011; Asness et al., 2012; Costa & Kwon, 2020). The critical idea of risk parity portfolio optimization is to find a good balance between different factors. If we closely look at the risk parity optimization equation (Costa & Kwon, 2020, e.g., See the equation (3)), the entropy-like constraint plays a critical role. Then we can balance the risk of each factor. Although we cannot clearly state a tight relationship between our balanced entropy acquisition and the risk parity strategy, both goals are similar. So it would be interesting to see the close relationship between our balanced entropy and risk parity strategy.

### A.24 MODEL ARCHITECTURES FOR OUR EXPERIMENTS

In this section, we describe model architectures what we have used in our experiments.

**Toy example - moon dataset**

```
BNN(
  (classifier): Sequential(
    (0): Linear(in_features=2, out_features=72, bias=True)
    (1): ReLU(inplace=True)
    (2): Linear(in_features=72, out_features=72, bias=True)
    (3): Dropout(p=0.2, inplace=False)
    (4): ReLU(inplace=True)
    (5): Linear(in_features=72, out_features=72, bias=True)
    (6): Dropout(p=0.2, inplace=False)
    (7): ReLU(inplace=True)
    (8): Linear(in_features=72, out_features=3, bias=False)
  )
)
```

**MNIST**

```
CNNBNN(
  (features): Sequential(
    (0): CNN2D(in_channel=1, out_channel=32, kernel_size=5,
               stride=1, dropout_p=0.5, apply_max_pool=True,
               apply_relu=True)
    (1): CNN2D(in_channel=32, out_channel=64, kernel_size=5,
               stride=1, dropout_p=0.5, apply_max_pool=True,
               apply_relu=True)
  )
  (classifier): Sequential(
    (0): Linear(in_features=1024, out_features=128, bias=True)
    (1): Dropout(p=0.5, inplace=False)
    (2): ReLU(inplace=True)
    (3): Linear(in_features=128, out_features=10, bias=False)
  )
)
```

**SVHN**

```
RESNETBNN(
  (features): ResNet18(remove_last_fully_connected_layer=True)
  (classifier): Sequential(
    (0): Linear(in_features=512, out_features=512, bias=True)
    (1): Dropout(p=0.2, inplace=False)
    (2): ReLU(inplace=True)
    (3): Linear(in_features=512, out_features=10, bias=False)
  )
)
```

**CIFAR-100 and $3\times$CIFAR-100**

```
RESNETCLASSIFIER(
  (classifier): Sequential(
    (0): Linear(in_features=2048, out_features=2048, bias=True)
    (1): Dropout(p=0.2, inplace=False)
    (2): ReLU(inplace=True)
    (3): Linear(in_features=2048, out_features=100, bias=False)
  )
)
```

**TinyImageNet**

```
RESNETBNN(
  (features): ResNet50(remove_last_fully_connected_layer=True)
  (classifier): Sequential(
    (0): Linear(in_features=2048, out_features=2048, bias=True)
    (1): Dropout(p=0.2, inplace=False)
    (2): ReLU(inplace=True)
    (3): Linear(in_features=2048, out_features=200, bias=False)
  )
)
```

