# OpenReview forum: "Active Learning in Bayesian Neural Networks with Balanced Entropy Learning Principle"
_ICLR.cc/2023/Conference — ICLR 2023 notable top 25%_

### Official Review · Reviewer_MCdP · 2022-10-23

**Confidence:** 4
**Correctness:** 3
**Technical Novelty And Significance:** 3
**Empirical Novelty And Significance:** 2
**Recommendation:** 6

**Clarity, Quality, Novelty And Reproducibility:**

The paper is generally well-written and easy to follow. The proposed acquisition function looks novel to me, and it is theoretically motivated. I suggest the authors address the above-mentioned weaknesses to improve the quality of the paper.

**Strength And Weaknesses:**

Strength:
1. This paper proposes a theoretically-motivated (Thm 4.1) acquisition function BalEntAcq that can select a diverse set of data points near the decision boundary (at least in the toy example studied in Section 4.2).
2. The acquisition function can be computed without relying on other data points, which allows parallelization in sampling/computation.
3. Empirical results show the efficacy of the proposed acquisition function.

Weaknesses:
1. It seems that the calculation of BalEnt relies on the Beta distribution approximation of the marginal probability. How difficult is it to calculate the acquisition function without the Beta distribution approximation?
2. It would be great if the authors can add further explanations for the choice of BalEnt[x]^{-1} in BalEntAcq. Currently, it seems to me that such a choice is mainly motivated by the empirical results shown in Appendix A.13.2.
3. Since one selling point of the proposed acquisition function is that it can be computed without relying on other data points, many existing methods (e.g., BADGE) do. I think the authors should add a discussion on the computational gain of the proposed method (e.g., what is the computational complexity of your proposed method, and what is the computational complexity of BADGE).

**Summary Of The Paper:**

This paper proposes a new acquisition function Balanced Entropy Acquisition (BalEntAcq) for active learning, which captures the information balance between the uncertainty of underlying softmax probability and the label distribution. The new acquisition is designed to be computed without relying on other data points, which allows parallelization in computation. With the Beta distribution approximation of the marginal distribution, the authors derive a closed-form expression of the proposed acquisition. The authors run experiments and show the efficacy of the proposed method.

**Summary Of The Review:**

The proposed acquisition function is theoretically motivated and allows parallelization in computation; empirical results also confirm the efficacy of the proposed method. Overall, I am inclined to accept this paper. However, I believe this paper can be made stronger if the authors can address the weaknesses mentioned above.

====after rebuttal====

I would like to thank the authors for their response. I have read the response and the comments from other reviewers, and I would like to remain my scores.

---

> ### Author Response · Authors · 2022-11-17
> **Thank you very much for your time and feedback.**
>
> Thank you very much for your valuable and highly constructive feedback.
>
> > It seems that the calculation of BalEnt relies on the Beta distribution approximation of the marginal probability. How difficult is it to calculate the acquisition function without the Beta distribution approximation?
> * Thank you for this excellent question. Beta distribution approximation facilitates several advantages: 1) BALD[x] is a function of its marginals, as proved in Theorem 3.1. and we use this lesson to justify the choice of the marginalized joint entropy MJEnt[x] as well, 2) simple two-parameter $(\alpha,\beta)$ estimation by matching the mean and variance of each marginal, and 3) closed-form expression in MJEnt[x] is possible. Therefore, the computational cost of BalEnt[x] is very cheap. Because of the marginalization, we don't have to worry about the dependency between variables $(P_1,\cdots, P_C)$. Moreover, the posterior distribution is automatic by the conjugate family of the Beta distribution (or Dirichlet distribution).
> * Therefore, without Beta distribution approximation, we CANNOT guarantee that 1) BALD[x] is a function of its marginals, and 2) the closed-form expression in MJEnt[x] exists. Then the acceptable choice is to apply the distribution-free estimation of the posterior distribution similar to CoreMSE, which requires much more expensive computational costs (see Table 8 in Appendix A.18). This implies that we cannot expect computational benefits once we avoid Beta distribution approximations.
>
> > It would be great if the authors can add further explanations for the choice of BalEnt[x]^{-1} in BalEntAcq. Currently, it seems to me that such a choice is mainly motivated by the empirical results shown in Appendix A.13.2.
> * As explained in response to other reviewers, we added further discussion on the explanation of our motivation to BalEnt in **Appendix A.20** on page 34. Our specific ratio is motivated by the application of Fano's inequality in the proof of Theorem 4.1 in Appendix A.5. Moreover, we added a comparison with the acquisition without the ratio form MJEntAcq[x] in the later part of Appendix A.20. Then, we confirm that our specific ratio form BalEnt[x] helped to improve the active learning performance.
>
> > Since one selling point of the proposed acquisition function is that it can be computed without relying on other data points, many existing methods (e.g., BADGE) do. I think the authors should add a discussion on the computational gain of the proposed method (e.g., what is the computational complexity of your proposed method, and what is the computational complexity of BADGE).
> * As explained in response to the reviewer VNzQ, we have updated **Appendix A.18** on page 32.
> * We added acquisition time measurement to the unlabelled dataset size $N$ and acquisition size $K$ for BalEntAcq and BADGE. As described in the complexity table (see Table 8), we confirm that both methods are linearly proportional to the unlabelled dataset size $N$ (See Figure 21 on page 33). We note that the computational time slope of BADGE is much steep. However, our BalEntAcq does not rely on the acquisition size $K$. Because of the dependency of the acquisition size $K$ in BADGE, it obstructs the parallelization.
>
> We would be happy to clarify further if you have other follow-up questions. We hope our response mitigates your concerns as much as possible. Thank you very much again.

---

> ### Author Response · Authors · 2022-12-08
> **Thank you very much.**
>
> We just checked the updated review response. Thank you very much again for the time and keeping the score as-is.

---

### Official Review · Reviewer_gjo6 · 2022-10-24

**Confidence:** 3
**Correctness:** 4
**Technical Novelty And Significance:** 3
**Empirical Novelty And Significance:** 3
**Recommendation:** 8

**Clarity, Quality, Novelty And Reproducibility:**

**Clarity**

The paper is quite well written and thorough with sufficient details and examples.

**Quality and Novelty**

To the best of my knowledge the proposed acquisition function appears to be novel. The paper is also well framed and results in significant empirical improvements.

**Reproducibility**

The authors provide code with the submission along with sufficient details in the paper to reproduce the results.

**Strength And Weaknesses:**

**Strengths**

- The paper is very clearly written and easy to follow. The systematic analysis of Bayesian NNs and existing approaches in active learning approaches is quite insightful.
- The proposed acquisition function is conceptually simple and can be easily adopted in practice. The toy experiments are also quite helpful in illustrating the effect of the acquisition function - selection of _diverse_ and _informative_ points.
- The acquisition function also enjoys strong empirical performance in realistic large datasets, performing on par with or better than more sophisticated and expensive-to-evaluate acquisition functions. The experimental analysis is also quite thorough and provides useful insights.

**Weaknesses**

- I believe the key weakness of the approach to me is that the acquisition function is stated directly in a form without much motivation. It would be nice to understand how the particular ratio comes and the motivations behind said choice.

**Summary Of The Paper:**

This paper tackles the problem of selecting the batches from unlabelled datasets to be labelled that are the most informative for a classifier trained on the data. This problem is traditionally studied within the paradigm of active learning, where the data to be labelled is collected sequentially in batches. The batches are selected using some notion of information gain to the current classifier. The paper presents a novel acquisition function for batch selection in active learning. The authors first examine the standard Bayesian Neural network paradigm used in active learning, and study the behavior of existing acquisition functions like BALD. The authors then outline the Balanced Entropy acquisition function, which is the ratio between the marginalized joint entropy and the entropy (plus a constant). The authors then study the behavior of this acquisition function theoretically as well as through a toy experiment. Finally the authors present an empirical study of their approach on large realistic datasets.

**Summary Of The Review:**

In summary, the paper presents an novel acquisition function for active learning resulting in significant improvements on realistic large datasets. The paper is well written and accompanied with thorough analysis. The motivation for the exact form of the acquisition function is somewhat lacking and would be useful to add. I lean towards acceptance.

---

> ### Author Response · Authors · 2022-11-17
> **Thank you very much for your time and feedback.**
>
> Thank you very much for your valuable and favorable feedback.
>
> > I believe the key weakness of the approach to me is that the acquisition function is stated directly in a form without much motivation. It would be nice to understand how the particular ratio comes and the motivations behind said choice.
> * As explained in response to the reviewer VNzQ, we added further discussion on the explanation of our motivation to BalEnt in **Appendix A.20** on page 34. From our perspective, the most challenging part to justify our formulation of BalEnt[x] is the choice of the precision level $-\log\Delta\approx H(Y)+\log 2$. On the one hand, the choice of MJEnt[x] in BalEnt[x] is natural by applying Jensen's inequality. On the other hand, since $H(Y)+\log 2$ is the minimally required nats (or bits) to encode $Y$ with auxiliary nats (or single bit) from the error entropy by Fano's inequality, we see that there exists the information limit near $H(Y)+\log 2$ when we obtain the label information.
> * Then aligning with matching MJEnt[x] in BalEnt[x], our BalEntAcq[x] performs well. Because of our limited understanding in the application of the point process entropy, we tried to demonstrate our choices empirically as much as possible, e.g., in Appendix A.13.2 and Appendix A.13.3.
> * The particular ratio form in BalEnt[x] is motivated by the proof leveraging Fano's inequality in Appendix A.5. However, to mitigate the concern about the ratio form in BalEnt[x], we added a comparison with the acquisition without the ratio form in Appendix A.20. We confirm that our particular ratio form help to improve the active learning performance.
>
> We appreciate your time and efforts in reviewing our paper again. We would be happy to clarify further if you have any additional questions.

---

> > ### Comment · Reviewer_gjo6 · 2022-11-18
> > **Thanks for the response!**
> >
> > Thank you for the response and taking my concerns into consideration.
> >
> > I think the response and additional details in Appendix A.20 address my concern and satisfactorily clarify the motivation for the chosen form for the acquisition function.
> >
> > I will keep my score and recommend acceptance. The work is quite thorough and presents a significant contribution to the field worth highlighting at the conference.

---

> > > ### Author Response · Authors · 2022-11-19
> > > **Thank you very much again.**
> > >
> > > We are so grateful for taking the time and positive feedback.

---

### Official Review · Reviewer_VNzQ · 2022-10-25

**Confidence:** 3
**Correctness:** 3
**Technical Novelty And Significance:** 3
**Empirical Novelty And Significance:** 3
**Recommendation:** 8

**Clarity, Quality, Novelty And Reproducibility:**

This paper provides a comprehensive review of relevant existing methods, providing a good context for the proposed work.
The technical details, motivation, and main contributions of the proposed uncertainty measure, BalEntAcq, are fairly clearly stated, and the expected merits of the resulting AL scheme are sufficiently supported by extensive performance assessment results.
The proposed scheme is reasonable and moderately novel, but its main strength would be its enhanced AL performance in a variety of practical settings and its low computational cost and scalability rather than novelty.



**Strength And Weaknesses:**

Strength:

Through extensive simulations and comparison with some of the existing popular active learning (AL) schemes, the paper demonstrates the potential benefits of the proposed uncertainty measure, BalEntAcq.
Especially, BalEntAcq has been shown to lead to the acquisition of diversified labels for data points in the pool that are well spread along the decision boundaries, thereby resulting in more efficient model enhancement through label acquisition.
Furthermore, the computational cost of the proposed strategy seems to be fairly low compared to many AL strategies based on Bayesian experimental design (BED), thanks to the approximation of the marginal distributions based on Beta distributions.
This makes the proposed AL method possibly much more scalable compared to the BED, which is an important potential merit of the proposed scheme.
Overall, the authors have shown that AL via BalEntAcq generally leads to performance improvement over other popular schemes based on several widely used benchmarks.

Weakness:

While scalability is one of the main advantages of the proposed method, direct performance assessment results demonstrating the scalability of BalEntAcq appears to be missing in the current study.
It would be interesting and important to see the comparison between the proposed method compared to other schemes, in terms of computational cost and scalability, especially with methods that the paper mentions to be too costly or not scalable (e.g., BADGE, CoreMSE, and other BED based schemes).
Furthermore, it would be helpful to provide experimental support for "linear scalability" and "exponential savings" that can be attained via AL using BalEntAcq as claimed in the current study.

The implications of the sign of BalEnt, the proximity of BalEnt to 0, and its relevance to the location of the corresponding data point with respect to the decision boundaries could (and should) be better explained, considering their importance in how BalEntAcq leads to enhancing the AL performance.



**Summary Of The Paper:**

This paper presents a novel acquisition function to improve active learning performance in Bayesian neural networks.
The acquisition function, referred to as Balanced Entropy Acquisition (BalEntAcq), serves as an uncertainty measure that aims to capture the information balance (or lack thereof) between the uncertainty of the underlying softmax probability computed by the Bayesian neural network and the label to be predicted.


**Summary Of The Review:**

This paper presents a new uncertainty measure, BalEntAcq, that has the potential to improve AL performance in a variety of settings while still being scalable (e.g., compared to other existing Bayesian schemes).
The authors support their claims based on extensive performance evaluation results using widely used benchmarks, which demonstrate the potential advantages of the proposed AL acquisition function.

---------

The scores have been updated after reviewing the authors' clarifications to the review comments.

---

> ### Author Response · Authors · 2022-11-17
> **Thank you very much for your time and feedback.**
>
> Thank you very much for your valuable feedback. In this response, we would like to explain our updates to address your concerns:
> > It would be interesting and important to see the comparison between the proposed method compared to other schemes, in terms of computational cost and scalability, especially with methods that the paper mentions to be too costly or not scalable (e.g., BADGE, CoreMSE, and other BED based schemes)
>
> * We have updated **Appendix A.18** on page 32 by adding CoreMSE and space complexity. As explained, CoreMSE shows quadratic time $O(C^2N^2K)$ and space complexity $O(C^2N^2)$ where $C$ is the number of classes, $N$ is the number of unlabelled images, and $K$ is the acquisition size. In reality, we cannot run CoreMSE properly because of memory deficiency. Therefore, the most interesting comparison, given scalability, is with BADGE.
> * We added acquisition time measurement to the unlabelled dataset size $N$ and acquisition size $K$ for BalEntAcq and BADGE. As described in the complexity table (see Table 8), we confirm that both methods are linearly proportional to the unlabelled dataset size $N$ (See Figure 21 on page 33). However, our BalEntAcq does not rely on the acquisition size. Because of the dependency of the acquisition size $K$ in BADGE, it blocks the parallelization.
>
> > Furthermore, it would be helpful to provide experimental support for "linear scalability" and "exponential savings" that can be attained via AL using BalEntAcq as claimed in the current study.
> * To demonstrate the potential of exponential savings, we tested the active learning performance when $\tau$-many redundant images exist where $\tau\in[1,3,5,7,10,20,50]$ with CIFAR-100 dataset. As suggested, we added a new experimental result in **Appendix A.19** on page 33. This new experiment suggests that our BalEntAcq selects the important dataset significantly. For example, with 50 redundant images (total of 2.5M images), we still can achieve the full supervised accuracy with around 30K images which are $\leq 1.25$% of the entire dataset.
>
> > The implications of the sign of BalEnt, the proximity of BalEnt to 0, and its relevance to the location of the corresponding data point with respect to the decision boundaries could (and should) be better explained, considering their importance in how BalEntAcq leads to enhancing the AL performance.
> * We added further discussion on the explanation of the selection priority with the proximity of BalEnt to 0 in **Appendix A.20** on page 34. As proved in Appendix A.5, BalEnt has a natural interpretation as a generalized estimation error probability. Then the positive probability implies that there exists a fundamental limit to estimate $P_i^+$ given $Y$. Filling the gap from the different proximity choices to BalEnt is explained in Appendix A.13.2.
> * The choice of the precision level $-\log\Delta\approx H(Y)+\log 2$ is closely related to MJEnt[x] from Jensen's inequality in Appendix A.2. Although we are not aware of the exact physical dynamics to describe MJEnt[x], Jensen's inequality still guides us that it is the maximally achievable state. Therefore, we see that MJEnt[x] should be more meaningful than any other choices of $-\log\Delta\$. Moreover, Appendix A13.3 suggests that our choice to match MJEnt[x] in BalEnt[x] is reasonably performing well.
>
> We would be happy to clarify further in this discussion thread if you have any other follow-up questions. We hope our new addition of appendix sections mitigates your concerns as much as possible. Thank you very much again.

---

> > ### Comment · Reviewer_VNzQ · 2022-11-17
> > **Thank you for the clarifications**
> >
> > The clarifications provided by the authors have been very helpful.
> > I believe the revised appendix and manuscript would strengthen the work further, and I will be happy to update the evaluation scores accordingly.
> > Thank you.

---

> > > ### Author Response · Authors · 2022-11-17
> > > **Thank you very much again.**
> > >
> > > Thank you very much again for taking the time and positive response.

---

### Author Response · Authors · 2022-11-16
**Thank you very much for your time in reviewing and favorably constructive feedback**

Dear reviewers,

We sincerely appreciate for taking the time to review our paper with positive and constructive feedback. We have updated the draft to accommodate your concerns as much as possible. Generally, two fundamental problems with the current version have been commonly raised. One is the need for a detailed discussion about scalability and exponential savings. The other is the need for more discussion on the motivation of our proposed BalEntAcq form. Therefore, we have revised the draft by adding more appendix sections, red colored in the updated version:
- We have improved **Appendix A.18** (previously A.16) on page 32 to discuss further both time and space complexity
- We added **Appendix A.19** on page 33 to explain the possibility of exponential savings.
- We added **Appendix A.20** on page 34 to explain the motivation of our BalEntAcq ratio form.

In our response to each reviewer, we shall include itemized answers for each concern. Please allow us **another day** to finalize our initial responses. Thank you again for your time and efforts.

Sincerely,

the Authors

---

> ### Author Response · Authors · 2022-11-17
> **The caption of Figure 23 in page 35 has been updated as of Nov. 17th.**
>
> We found a misprint in the caption of Figure 23. So this is to note our update in the draft. Thank you very much.

---

### Decision · Program_Chairs · 2023-01-20

**Decision:**

Accept: notable-top-25%

**Justification For Why Not Higher Score:**

The paper presents a good contribution to the area of active learning. The results obtained are good and promising, but at the moment there is a large number of different acquisition functions presented in this area and most of them only provide marginal gains. This paper is no different.

**Justification For Why Not Lower Score:**

All reviewers vote for acceptance. 2 with strong scores.

**Metareview: Summary, Strengths And Weaknesses:**

Summary:

This paper proposes a new acquisition function Balanced Entropy Acquisition (BalEntAcq) for active learning, which captures the information balance between the uncertainty of underlying softmax probability and the label distribution.  The new acquisition is designed to be computed without relying on other data points, which allows parallelization in computation. With the Beta distribution approximation of the marginal distribution, the authors derive a closed-form expression of the proposed acquisition. The authors run experiments and show the efficacy of the proposed method.

Strengths:

- Through extensive simulations and comparisons with baselines.
- Comprehensive review of relevant existing methods
- Enhanced AL performance in a variety of practical settings and its low computational cost and scalability.
- Clearly written and easy to follow.
- The acquisition function can be computed without relying on other data points, which allows parallelization in sampling/computation.

Weaknesses:

- Scalability results not clear.
- The proposed acquisition function could be motivated better.

Recommendation:

All the reviewers vote for acceptance. I, therefore, recommend to accept the paper and encourage the authors to use the feedback provided to improve the paper for its final version.


**Note From Pc:**

if the above contains the word "oral" or "spotlight" please see: "oral" presentation means -> notable-top-5% and "spotlight" means -> notable-top-25%. As stated in our emails, we are disassociating presentation type from AC recommendations